# Chimpanzee groups achieve sustainable resource use in a common-pool resource dilemma
Kirsten Sutherland [1] ✉, Daniel Haun [1,3] & Alejandro Sánchez-Amaro[1,2,3]

Common-pool resource dilemmas are group resource sustainability problems that are sensitive to over-extraction. While human strategies for overcoming common-pool resource dilemmas are well studied, the comparative evolutionary perspective has received little attention. Here, we compare resource management of chimpanzees (*N* = 15) grouped as dyads and quartets using an original experimental paradigm. The participants could use sticks to feed from a pool of yoghurt. The number of sticks equalled the number of players, and removing all of the sticks triggered resource collapse, thereby creating a social dilemma. Quartets were found to maintain the resource longer than dyads. Quartets', but not dyads', success was positively associated with social tolerance. Furthermore, quartets were more successful when the dominant ape acquired the relative lowest payoff. These results suggest that chimpanzees respond differently to cooperative sustainability problems depending on group size, with social tolerance playing an important role. The findings have implications for studying the evolution and diversity of hominid cooperation, in particular, highlighting that group size should be carefully considered in the design of non-human primate cooperation experiments.

Common-pool resource dilemmas are characterised by situations where a depletable resource is openly accessible[1]. In these scenarios, for each individual, it is most beneficial in the short term to maximise their own extraction of the resource. However, at the group level, if every participant maximises their consumption, the resource will be exhausted, resulting in a long-term shared "tragedy". The archetypal example of a common-pool resource dilemma is a common grazing pasture. If all farmers keep as many livestock as possible on the pasture, it will be destroyed and fail to regenerate in the future[2]. Thus, a resource that could have been sustained for many years to support many farmers (a large, long-term, shared payoff) is instead used up quickly by relatively few (a small, short-term, individual payoff), to the detriment of everyone.

While the "tragedy of the commons" is classically presented as an inevitability[3], since the mid-1970s, field research has identified that large common-pool resource dilemmas *can* be resolved, and that success is more likely when an emphasis is placed on local management, self-organisation, equitable payoffs, and rules and punishments relating to resource extraction[1,4,5]. In the 1980s and early 1990s, researchers began integrating economic theory with experimental methods more rigorously[6]. This shift laid the foundations for today's rich body of laboratory work on adult human commons management. Under experimental conditions, different aspects of common-pool resource dilemmas can be isolated, in order to tease apart what helps and hinders human cooperation in these situations. Experimental work has largely focused on the impact of individual differences, including differing social motives[7,8], task structures, such as the tension between reward and punishment to yield optimal sustainability[9], and situational factors, including social variables such as power[10,11], communication[12–15] and group size[16–18]. The experimental work is generally in agreement with the field work, that communication, rules, punishment, and equity enhance common-pool resource sustainability in humans[19,20] though, it is clear that this challenging dilemma tests the limits of humans' exceptional cooperative abilities.

Despite a wealth of experimental work on common-pool resource dilemmas with adult humans, comparative and developmental perspectives remain largely understudied. In what we believe to be the only experimental studies with five-year-old children and non-human primates, Koomen and Herrmann[21,22] find that their common-pool resource dilemma was resolved by 40% of dyads of German six-year-olds. In keeping with the conventional adult findings, the dyads of children that were able to sustain the resource for the longest were pairs that communicated and established rules, and

[1]Max Planck Institute for Evolutionary Anthropology, Leipzig, Germany. [2]University of Stirling, Stirling, UK. [3]These authors contributed equally: Daniel Haun, Alejandro Sánchez-Amaro. ✉e-mail: kirstenasutherland@gmail.com

produced more equitable payoffs for both players. This is in stark contrast with chimpanzees (*Pan troglodytes*), who solved the dilemma at comparable rates, but with more successful dyads having highly unequal payoffs[22]. The most successful dyads of chimpanzees also had higher degrees of dominance asymmetry, mediated by a lower level of social tolerance, and so the highly disproportionate ratio of payoffs resulted from the more dominant individual monopolising the resource. This finding is congruent with Schneider et al.'s[23] study of chimpanzees in a collective action problem. Here, rather than resisting an individual short-term benefit in order to prevent a long-term group-level tragedy, the participants had to pay an individual cost in order to produce a shared group-level benefit. When group size was small and the resource was monopolisable, higher-rankers were more likely to pay the cost, as they were guaranteed a share of the reward. In scenarios that required cooperative action and when group size was larger, propensity to act was not predicted by rank.

These findings combine to produce a picture of chimpanzees as individualists constrained by social dynamics. If all participants act only out of economic self-interest, the "Tragedy of the Commons" should be an unavoidable outcome when presented with a common-pool resource dilemma. However, Koomen and Herrmann[22] find that chimpanzees use dominance and tolerance dynamics to avoid resource collapse when extracting from a common-pool resource in pairs, resulting in unequal payoffs. Presumably, given the opportunity, both participants would *like* to extract as much as possible from the pool; however, in practice, only particularly dominant individuals are able to.

This presents several open questions on the topic of common-pool resource management from the comparative psychology perspective. The common-pool resource dilemma is theoretically a group-level problem, and arguably, dyads are not simply small groups[24]. The vast majority of the human work on the topic of the commons focuses on groups of various sizes greater than two[20]. To truly test the common-pool resource management abilities of non-human great apes, both for the sake of better understanding their behaviour and to embed human behaviour in its evolutionary context, participation in group-level tests is needed.

Within the existing non-human great ape experimental cooperation and economic behaviour literature, there is a striking absence of group-level studies[25,26]. This represents a serious gap in our understanding of how great apes, as group-living animals with highly complex social groups, cooperate. Despite recent efforts to correct this shortcoming, the current state of the field remains fragmented. In addition to the previously described collective action study by Schneider et al.[23], we have identified two captive experimental studies of chimpanzees on the topic of cooperation and social dilemmas that include more than 2 participants. In a triadic ultimatum game where the responder could choose an ultimatum offered by two competing proposers, Sánchez-Amaro et al.[27] find that chimpanzee proposers competitively increase their offers over trials. In a task where triads and dyads of chimpanzees had to coordinate simultaneous actions to obtain rewards, Suchak et al.[28] report a high level of success, with the level of cooperation generally improving over time, though group-size differences are not discussed.

With this in mind, the aims of the current study are threefold: firstly, to examine the effect of group size (dyad versus quartet) on chimpanzees in a common-pool resource dilemma. Secondly, to further investigate the positive effect of dominance-asymmetries and low social tolerance on common-pool resource sustainability reported by Koomen and Herrmann[22], and its possible interaction with group size and thirdly, to test this experimentally with an intuitive apparatus and simple paradigm. By developing an apparatus that the majority of participants could use without training, we hoped to maximise the size and representativeness of the participant pool (see Supplementary Information Section 2.2 for pre-experience details and comprehension criteria).

The apparatus (Fig. 1) consists of an elevated pool that contains 1 kg of plain 10% fat yoghurt. The participants were able to dip into the yoghurt using sticks. The sticks had a dual function of being both the tool that the apes used to extract and consume yoghurt, and holding up a transparent lid

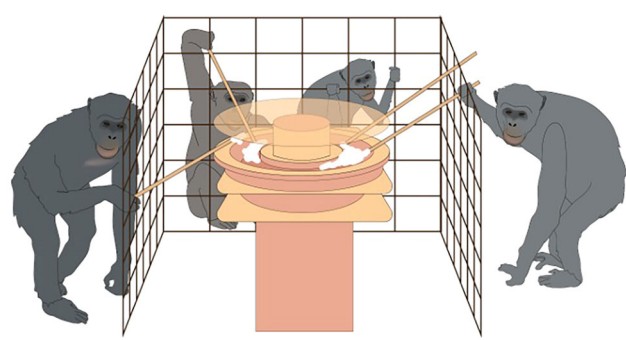

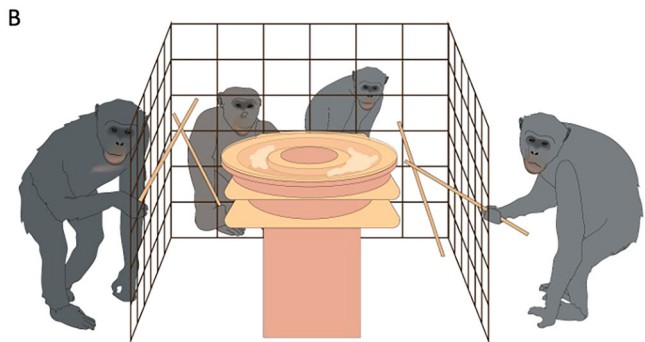

**Fig. 1 | Illustration of experiment setup. A** Quartet participation with one player sitting out. One stick remains supporting the lid and preventing collapse. **B** Resource collapse after all sticks are removed.

that covered the apparatus. When unsupported, the lid would slowly start to close. From the start position, the apes had a total of 10 seconds to intervene and reinsert a stick to prevent the collapse. Once the lid had fully closed, it could not be reopened, and the trial was over. Full closure represented resource collapse.

The number of sticks provided was always equal to the number of players. This created a Common-Pool-Resource dilemma dynamic where the sticks were subtractable and openly accessible, but if all of the sticks were removed, it would result in a collective negative outcome. We used the latency between the start of the test (the moment the doors to the testing room opened) and the moment of resource collapse as a measure of success in resource sustainability. There was also a control condition with no lid, where the group was able to remove all of the sticks with no negative consequences.

Four hypotheses were constructed to address the study aims and were pre-registered on OSF (https://osf.io/h7245). Deviations from the pre-registration are reported in Section 1 of the Supplementary Information. Firstly, we hypothesised that quartets would have more difficulty than dyads in sustaining the common-pool resource, resulting in shorter collapse latencies. Chimpanzee dyads have been found to be less able to sustain resources when playing with competitors compared to when playing alone[22] though, to our knowledge, groups larger than dyads and other species of non-human great ape have not yet been studied. Large groups of humans do succeed in maintaining common-pool resources; however, local-level organisation and an emphasis on fairness are usually required[29]. As there is no evidence for non-human apes developing rules to mediate resource division or facilitate coordination[30–32], and because we expected dominant individuals to be less able to monopolise the resource in quartets[22,23], it was hypothesised that quartets would sustain the resource for less time than dyads.

Hypothesis 2 stated that all sticks would be removed from the common pool more quickly in the absence of the social dilemma (i.e., in the control condition). This was measured in the latency between the start of the trial (the moment the door to the testing room opened) and the moment the last remaining stick was picked up. The measure aimed to distinguish between conservative use of the resource resulting purely from low cofeeding tolerance[33] and conservative use resulting from the social dilemma. For example, if a group contained one participant who consistently waited for over two minutes before taking a stick, while the other group member(s) took sticks within ten seconds, it would be important to determine whether the slower participant was prevented from approaching by low cofeeding tolerance. In this case, their latencies for removing sticks should be no different to their latencies in the control condition. If their behaviour had a more strategic motivation, where they refrained from removing a stick to avoid resource collapse, we would expect to see shorter stick-removal latencies in the control compared to the test condition, as the participant would have no reason not to approach and remove a stick in the control. We did not expect the chimpanzees to develop turn-taking or stick-sharing strategies as humans might. Therefore, the most likely solution would be for them to leave one stick inside the pool to keep it open. The last stick-removal latency was therefore of interest, as it was expected to trigger resource collapse.

Hypothesis 3 stated that groups with heterogeneous dominance and low social tolerance will be more successful at sustaining common-pool resources than those with similarly matched dominance rankings, and that this effect would be greater in dyads. This hypothesis is in line with Koomen and Herrmann's[22] finding that chimpanzee pairs most successfully sustain common-pool resources in dyads that are characterised by low social tolerance, asymmetrical dominance, and, consequently, asymmetrical payoffs. It is unclear how adding players to the dilemma may alter this dynamic, as very little research has been conducted on group-level social dilemmas in great apes; however, we hypothesised that the effect will be greater in dyads, as it should be more difficult for a dominant individual to monitor the behaviour of three other players compared to one[23].

Like hypothesis 3, hypothesis 4 was related to the prediction that success will be achieved through monopolisation by dominant individuals, and that this would be more difficult to achieve in larger groups. Hypothesis 4 stated that payoffs would be more unequal in dyadic groups than in quartets, especially in dyads that have asymmetrical dominance and low social tolerance. Note that hypothesis 4 was numbered 6 in the pre-registration. The hypotheses that were pre-registered as hypotheses 4 and 5 are presented in the SI.

## Methods

### Study site and subjects

Participants were housed in two separate enclosures (the A-group and the B-group) at the Wolfgang Köhler Primate Research Centre at Leipzig Zoo, Germany. 10 (8 females) of the 19 A-group chimpanzees and five (four females) of the six B-group chimpanzees passed the comprehension criteria (see Supplementary Information for training procedure) and were included in the study. Participants ranged in age from 7 to 48 years (mean = 28.9 years, $SD$ = 12.06). These 15 participants were organised into 24 testing groups (7 quartets and 17 dyads). Subgroups were voluntarily separated from the main group for testing and were returned to the main enclosure if there were signs of stress. Between February 2022 and April 2023, 708 trials were conducted, with four dropped (see Supplementary Information Section 2.4).

### Materials

The apparatus ($50 \times 50 \times 58$ cm) was made from 3D printed opaque plastic, apart from the lid, which was a 50 cm diameter circle cut from transparent plastic. The apparatus was placed in a booth in the testing room, where the apes could access it through mesh panels making up three sides of a rectangle (right and left panels measuring $48.5 \times 69$ cm, the middle panel measuring $82 \times 69$ cm). In the starting position, the lid was held up by wooden sticks (74 cm in length). The number of sticks was equal to the number of players. If all of the sticks were removed, the lid would slowly close, preventing access. The central pool was filled with approximately 1 kg of 10% fat plain yoghurt. The yoghurt could only be accessed by removing a stick and using it to dip. A compartment in the middle of the apparatus was filled with water. Holes in the base of the lid caused the lid to slowly fill with water, weighing it down. This caused the lid to close slowly (10 s in total) when unsupported. In the control condition, the lid was not used. Tests were recorded from four angles using four camcorders.

### Test procedure

Participants were called into testing rooms that adjoin their main enclosure by a zookeeper. Using commands and food rewards to move chimpanzees between compartments, the participants could be separated from the non-participants. When all of the non-participants had returned to the main enclosure, the groups of either two or four participants were moved to a "waiting" room connected to the testing room. The apparatus setup was then completed in the testing room. The trial began when the door between the waiting room and the testing room was opened. The apes could move freely between the two rooms throughout the test.

In a test-condition trial, the trial ended when the lid collapsed or when a cutoff was imposed (see Supplementary Information Section 2.5 for criteria). In a control-condition trial, the trial was over when all sticks were removed. As there was no collapse mechanism in the control condition, after the last stick was removed, the apes were left to freely eat the yoghurt for a minimum of one minute (though it was usually longer). The time that the last stick was removed was also recorded in the test condition, allowing trial latency comparisons between conditions. For analyses that only included test condition trials, the true collapse latency could be used. In the test condition, *Last Stick Latency* could be used to indicate the time of collapse as it was very strongly correlated with *Collapse Latency* ($r_s$ = 0.98, 95% CI [0.97, 0.99], $p$ = <0.001, $n$ = 374).

Based on a preliminary power analysis, 18 test trials and 18 control trials were planned for each of the 24 test groups, though seven groups did not complete all of their planned trials due to at least one participant declining to enter the testing room. Every test session, a randomised number of trials (between 1 and 5) was administered in a random combination of test and control conditions. It was therefore difficult for the participants to predict if a trial was their last attempt of the day, which should have emphasised resource collapse as an undesirable outcome. If a trial was the last of the day, the group would be released into their enclosure shortly after the trial finished. If they were to participate in another trial, they would be returned to the "waiting room", the apparatus would be refilled with yoghurt, returning the volume to approximately 1 kg, and the procedure would be repeated.

### Group dominance-difference

Four keepers at the Wolfgang Köhler Primate Research Centre were asked to rank the 10 A-group participants, and three keepers were asked to rank five B-group participants in order of dominance (such as in ref. 34). Kendall's coefficient of concordance found strong agreement between the four A-group raters ($W$ = 0.86, $\chi^2(8)$ = 27.6, $p$ = 0.001) and between the three B-group raters ($W$ = 0.96, $\chi^2(4)$ = 11.5, $p$ = 0.022). The mean of all ratings for an individual was taken as that individual's relative dominance ranking. Among the dyads, the group dominance-difference was simply the difference in ranking of the two individuals. For the quartets, the difference between every possible dyad configuration within the quartet was averaged (mean) in order to produce a group-level quantification of dominance asymmetry.

While our focus on dominance and tolerance dynamics draws from Koomen and Herrmann[22], we use different measures to investigate them. Koomen and Herrmann[22] (Study 2) established dominance and tolerance scores through a cofeeding pretest. Conducting this test was not possible in our zoo setting, so alternative methods were used. As a result, "tolerance" in this study is based on individuals' tendency to spend time in proximity to each other, not based on their cofeeding propensity in a cofeeding tolerance test (e.g., ref. 35).

## Group social tolerance

The term "social tolerance" is used inconsistently in the primate literature, but is generally defined as an absence of agonistic interactions[36]. Here, we use proximity, a form of behavioural social tolerance in which the presence of a conspecific is tolerated at a close distance[36,37], as a measure of social tolerance.

Proximity scans were conducted at the zoo enclosures in the early afternoon on every workday (Monday–Friday). For each individual, the identities of individuals within arm's reach were recorded. Research assistants used a tablet to record their observations with the behavioural coding software ZooMonitor[38].

For each testing group, the proximity scans that corresponded to the group's testing date range were isolated to account for changing relationships in the populations[39]. The simple dyadic association index from Cairns and Schwager[40] was selected as it is appropriate for a captive zoo setting where all animals are always visible. The following formula was used:

$$DAI = \frac{x}{x + Y_{AB}} \tag{1}$$

where $x$ is the number of sampling periods where $A$ and $B$ were associated, and $yAB$ is the number of sampling periods where $A$ and $B$ were identified but not associated[40]. For quartets, the index was calculated for every possible dyad in the group, and the mean was taken as the average group level of social tolerance. For dyads, the dyadic association index was taken.

## Payoff equality calculation

Payoffs were coded from video as raw counts of scoops of yoghurt consumed by each individual in each test. Payoff equality of each trial was represented as an entropy score using the Shannon entropy formula[41]. The entropy of a random variable quantifies the average level of uncertainty related to the variable's potential outcomes using the formula:

$$H = -\sum \left( \frac{count_i}{\sum count} \right) \log \left( \frac{count_i}{\sum count} \right) \tag{2}$$

where $H$ is entropy, and $i$ is the index for each individual in the testing group. After entropy was calculated for each trail, the values were normalised by dividing them by the logarithm of the group size, thereby scaling them relative to the potential maximum entropy for that group. High entropy indicates uncertainty about which individual ate a particular scoop, and therefore, higher-payoff equality. Zero entropy indicates complete certainty that one individual ate every scoop in a trial, therefore indicating low payoff equality.

## Analysis

A simulation-based power analysis was used to determine the number of trials to be completed for the available number of groups. All analyses were conducted using R (version 4.4.1). The brms package was used for models and the ggplot2 package for visualisations. Posterior distributions were approximated using Markov chain Monte Carlo (MCMC) sampling implemented via Stan. Four chains were run in parallel with default settings of 2000 iterations per chain (including 1000 warmup), resulting in 4000 post-warmup samples for inference. Models are summarised in Table 1.

Independent (fixed effect predictors and control) and dependent/outcome variables are in *italics*. Model 1 was a Bayesian general linear mixed model (GLMM) fitted to only test condition trials ($N = 374$) to address hypothesis 1. Model 1 tested the effect of the predictors *Group Size* (centred around 0), *Session Number* (and its interaction with *Group Size*), *Group Dominance Difference* (and its interaction with *Group Size*), and *Group Social Tolerance* (and its interaction with *Group Size*) on the gamma-distributed (log-link) outcome variable *Collapse Latency* (the time in seconds between the start of the trial and the lid closure, representing resource collapse). The shape parameter of the Gamma distribution was modelled as a function of *Group Size*, allowing skewness and variability of *Collapse*

*Latency* to vary across group sizes. Random intercepts for *Group ID* were added, accounting for variability between groups, and random slopes for *Session* allowed the effect of session to vary across groups. In addition to *Group ID*, Random intercepts were added for the names of players to account for individual random effects.

The prior for the intercept was set as normal with a mean of 2 and a standard deviation of 2. This moderately informative prior reflected a belief about the possible range of *Collapse Latency* that can vary flexibly, with the Gamma response distribution ensuring that predicted values were positive. For the shape parameter of the Gamma distribution, the prior for the intercept was set as normal with a mean of 2 and a standard deviation of 2. For *Group Size*, the prior was normal and weakly informative, with a mean of 0 and standard deviation of 1, allowing for more flexibility. The same prior was set for the coefficient of *Group Size* on the shape parameter. Relatively small effect sizes were anticipated for *Session, Group Dominance Difference* and *Tolerance*, so the prior was normal, with a mean of 0 and a standard deviation of 0.5. For the random effect standard deviations, an exponential prior with a rate parameter of 5 was used. This reflects a prior belief that variability between groups is generally small but can extend to larger values if supported by the data.

Models 2 and 3 used the same model structure and priors, swapping *Group Size* for *Condition* (centred around 0) and swapping *Collapse Latency* for the gamma-distributed (log-link) outcome variable *Last Stick Latency* (the time in seconds between the start of the trial and the last stick being removed from the resource). In order to address hypothesis 2, Model 2 was fitted to the dyad data (test and control trials) ($N = 500$), and model 3 was fitted to the quartet data (test and control trials) ($N = 204$).

Like Model 1, Model 4 is a Bayesian GLMM with the gamma-distributed (log-link) response variable *Collapse Latency*. Model 4 was formed post-hoc as an exploratory analysis and so does not correspond to any hypothesis; however, as it elaborates on the findings of Model 3, it is associated with hypothesis 2. Predictors are *Proportion* (centred), representing the proportion of time sticks were held by the dominant ape in the group relative to the other participants, *Group Dominance Difference* (centred), and *Group Social Tolerance* (centred). Stick-holding duration was used to calculate the proportion measure instead of absolute counts of yoghurt-mouthfuls eaten, as it accounted for individual differences in speed and dexterity. Random intercepts for *Group ID* and the name of individual players were added, accounting for variability between groups and players. The prior for the intercept was tightened to improve fit due to the complexity of the model. It was normally distributed, with a mean of 5 and a standard deviation of 0.3. As in the previous models, the effect of *Group Dominance Difference* and *Group Social Tolerance* was expected to be modest, so the prior was set as a mean of 0, normally distributed, with a standard deviation of 0.5. The same prior was used for the shape parameter. For the random effect standard deviations, an exponential prior with a rate parameter of 5 was used. Model 4 was applied to the quartet test-condition trials ($N = 101$). This was after five trials were removed from the analysis because accurate stick-holding durations could not be coded for at least one individual based on the available video footage.

Model 5 was applied to dyad test-condition trials ($N = 267$), after one trial was removed due to difficulty coding stick-holding durations from the available video footage. Model 5 used the same model structure and priors as Model 4, with an added fixed effect for the quadratic term for *Proportion* (centred). This controlled for the expected quadratic relationship between *Collapse Latency* and *Proportion* that would emerge uniquely in dyads. Because removing the last stick has a direct causal relationship with the lid closing, and in dyads, there are only two sticks, *Collapse Latency* should be high when *Proportion* is both low and high (see Fig. S2). By controlling for this, we could investigate whether there is a stronger effect on latency when the *Proportion* is low or high.

Model 6 tested the relationship between *Collapse Latency, Group Size*, and their interaction on payoff equality, calculated as *Normalised Entropy* scores in test trials ($N = 374$). The Zero-One Inflated Beta response family was chosen as *Normalised Entropy* is bound between 0 and 1, with some

**Table 1 | Summary of the purpose of statistical models**

| Hypothesis | Model | Model aim | Model formula |
|---|---|---|---|
| H1 - quartets will have more difficulty than dyads in sustaining the common-pool resource, resulting in shorter collapse latencies. | Model 1 | Applied to test condition trials. Tests the effect of group size on collapse latency (time from the start when the lid touches the bottom). | *Collapse Latency ~ 1 + Group Size * Session Number + Group Size * Group Dominance Difference + Group Size * Group Social Tolerance + Group Dominance Difference * Group Social Tolerance + (1 + Session Number\|Group ID) + (1\|Player Name)* <br> *Shape ~ 1 + Group Size* |
| H2 - participants will approach and extract from the common pool more quickly in the absence of the social dilemma (i.e., in the control condition). | Model 2 | Applied to dyad trials. Tests the effect of the condition (test or control) on the time until all sticks are removed from the resource. | *Last Stick Latency ~ 1 + Coundition * Session Number + Condition * Group Dominance Difference + Condition * Group Social Tolerance + Group Dominance Difference * Group Social Tolerance + (1 + Condition + Session Number\|Group ID) + (1\|Player name)* <br> *Shape ~ 1 + Condition* |
| | Model 3 | Applied to quartet trials (otherwise, the same as Model 2). Tests the effect of the condition (test or control) on the time until all sticks are removed from the resource. | *Last Stick Latency ~ 1 + Coundition * Session Number + Condition * Group Dominance Difference + Condition * Group Social Tolerance + Group Dominance Difference * Group Social Tolerance + (1 + Condition + Session Number\|Group ID) + (1\|Player name)* <br> *Shape ~ 1 + Condition* |
| H3 - groups with heterogeneous dominance and low social tolerance will be more successful than those with similarly matched dominance rankings. This effect will be greater in dyads. | Model 2 | Applied to dyad trials. Tests the interacting effects of dominance and tolerance on the time until all sticks are removed from the resource. | *Last Stick Latency ~ 1 + Coundition * Session Number + Condition * Group Dominance Difference + Condition * Group Social Tolerance + Group Dominance Difference * Group Social Tolerance + (1 + Condition + Session Number\|Group ID) + (1\|Player name)* <br> *Shape ~ 1 + Condition* |
| | Model 3 | Applied to quartet trials. Tests the interacting effects of dominance and tolerance on the time until all sticks are removed from the resource. | *Last Stick Latency ~ 1 + Coundition * Session Number + Condition * Group Dominance Difference + Condition * Group Social Tolerance + Group Dominance Difference * Group Social Tolerance + (1 + Condition + Session Number\|Group ID) + (1\|Player name)* <br> *Shape ~ 1 + Condition* |
| | Model 4 | Applied to the quartet test trials. Further explores the findings of Model 3. Tests the effect of the proportion of resource use by the highest-ranker, and its interaction with dominance and tolerance on time until resource collapse. | *Collapse Latency ~ 1 + Proportion * Group Dominance Difference * Group Social Tolerance + (1\|Group ID) + (1\|Player name)* |
| | Model 5 | Applied to dyad test trials. Fits Model 4 to the dyad dataset to explore whether the *proportion* of resource use by the highest-ranker has an effect. | *Collapse Latency ~ 1 + Proportion * Group Dominance Difference * Group Social Tolerance + Proportion$^2$ + (1\|Group ID) + (1\|Player name)* |
| H4 - payoffs will be more unequal in dyadic groups than in quartets, and this effect would be enhanced in groups that had asymmetrical dominance. | Model 6 | Applied to test trials. Addresses the first part of H4 by testing the interacting effect of collapse latency and group size on payoff equality. | *Normalised Entropy ~ log(Collapse Latency) + Group Size + (1\|Group ID) + (1\|Player name)* |
| | Model 7 | Applied to dyadic test conditions. Addresses the second part of H4 by testing whether payoff equality is predicted by dominance, tolerance, or their interactions in the dyad dataset. | *Normalised Entropy ~ 1 + Group Dominance Difference + Group Social Tolerance + (1\|Group ID) + (1\|Player name)* |
| | Model 8 | Applied to the quartet test conditions. Addresses the second part of H4 by testing whether payoff equality is predicted by dominance, tolerance, or their interactions in the dyad dataset. | *Normalised Entropy ~ 1 + Group Dominance Difference + Group Social Tolerance + (1\|Group ID) + (1\|Player name)* |

values of exactly 0 and 1 in the dyad subset. Normal and weakly informative priors were chosen for the fixed effects and the intercept, with means of 0 and standard deviations of 1.

Models 7 and 8 were fitted to the dyad data ($N = 268$) and the quartet ($N = 106$) data, respectively, using only the test condition trials. They tested the interacting effects of *Dominance Difference* and *Group Social Tolerance* on *Normalised Entropy*. Like with Model 6, a Zero-One Inflated Beta response family was used. Normal and weakly informative priors were chosen for the fixed effects and the intercept, with means of 0 and standard deviations of 1. For *Dominance Difference* and *Group Social Tolerance,* the prior was set as a mean of 0, normally distributed, with a standard deviation of 0.5. For the random effect standard deviations, an exponential prior with a rate parameter of 5 was used.

## Ethics statement

The study was reviewed and approved by an ethics committee from the Max Planck Institute for Evolutionary Anthropology and the Leipzig Zoo. The procedures used comply with the Weatherfall report 'The use of non-human primates in research' as well as with the EAZA Minimum Standards for the Accommodation and Care of Animals in Zoos and Aquaria, the WAZA Ethical Guidelines for the Conduct of Research on Animals by Zoos and Aquariums, and the ASAB/ABS's Guidelines for the Treatment of Animals in Behavioural Research and Teaching. IACUC approval was not necessary to conduct this research. Testing was always voluntary; the individuals were not food or water-deprived, and a trained animal caretaker did the handling. Returning to their social group was possible at any moment of the test, as well as terminating the test, given signs of discomfort

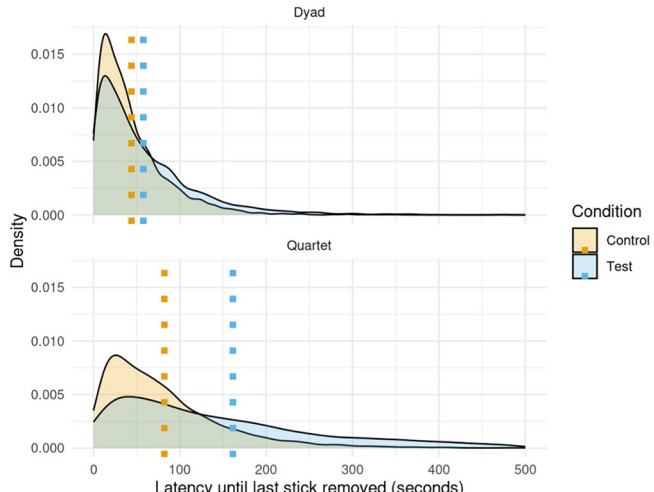

**Fig. 2 |** Density plot showing distribution of *Last Stick Latency* in dyads (*N* = 500 trials) compared to quartets (*N* = 204 trials), with mean latency for the control and test conditions marked.

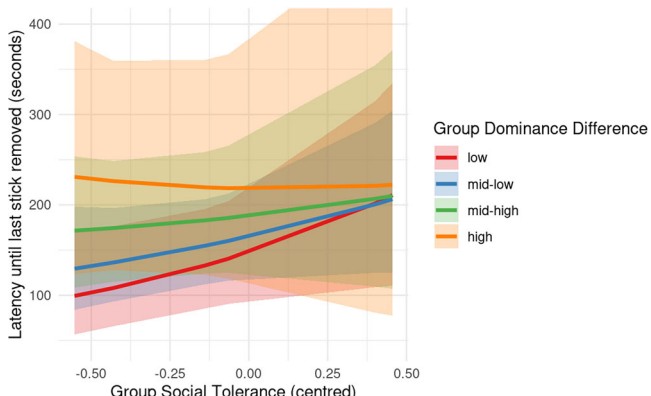

**Fig. 3 |** Interaction between *Group Dominance Difference* (centred) and *Group Social Tolerance* (centred) on quartet mean *Last Stick Latency* (s) in Model 3 (*N* = 204 trials). The shading shows a 95% credible interval around the predicted mean latency.

or distress. We did not separate infants from their mothers. We have complied with all relevant ethical regulations for animal use.

## Results

14% of trials were double-coded for reliability. Inter-rater reliability for holding times, total trial latency, and count payoffs was very high. Details are reported in the Supplementary Information.

### Group size

Our first pre-registered GLMM (Model 1) investigated the relationship between *Group Size* and *Collapse Latency* (the time between the start of the test and the moment the lid closed) in the test condition. We found that quartets sustained the common-pool resource for longer, for a mean of 168.0 s with a 95% Credible Interval (CI) of [103.0, 252.0], compared to a mean of 70.4 s with a 95% CI of [52.1, 96.0] in the dyads. A full posterior predictive contrast incorporating outcome-level variance found an average difference of 94 s (95% CI [31.1, 184], pd% = 100). This was contra to our hypothesis (H1) that *Collapse Latency* would be longer in dyads.

Additionally, we found substantial differences between the test and the control condition, evidencing that the chimpanzees adjusted their behaviour in response to the social dilemma. This was tested by a second and third pre-registered GLMM (Models 2 and 3) examining Hypothesis 2, that participants would remove all sticks from the Common Pool Resource more quickly in the absence of a social dilemma, measured by *Last Stick Latency* (the time between the start of the test, and the moment that the last remaining stick was removed from the pool) (Fig. 2). We found that in the quartets, the last stick was removed after a mean of 165.0 s in the test condition with a 95% CI of [119.0, 218.0], compared to a mean of 82.0 s in the control condition, with a 95% CI of [59.0, 108.0]. A full posterior predictive contrast incorporating outcome-level variance found an average difference of 83.1 s (95% CI [43.3, 132], pd% = 100).

In the dyads, latency until the last stick was removed was also longer in the test condition compared to the control, with a mean of 56.3 s and a 95% CI of [36.8, 82.5], in comparison to a mean of 43.8 s with a 95% CI of [27.0, 67.7]. The model slightly lacked statistical power to form firm conclusions based on such a small difference (12.5 s), and the 95% CI of the contrast slightly included zero ([−4.2, 28.8], pd% = 94). A full posterior predictive contrast incorporating outcome-level variance found an average difference of 12.3 s (95% CI [−4.6, 29.5], pd% = 93).

**Dominance and tolerance.** Using group-level composite measures of dominance-difference and social tolerance, we examined the effect of the

interaction between these two variables on *Last Stick Latency* (Hypothesis 3, Models 2 and 3). It was found that more tolerant quartets sustained the resource for longer. The main effect of *Group Social Tolerance* had a positive effect on latency in Model 3, with the most tolerant groups sustaining the resource on average 67 s longer than the least tolerant groups (95% CI [−5.5, 137.0], pd% = 97). A full posterior predictive contrast incorporating outcome-level variance found an average difference of 66.9 s (95% CI [−5.92, 137], pd% = 97).

While tolerance had an overall positive effect on resource sustainability among quartets, there was evidence that this positive effect was reduced as dominance rank became more unequal (Fig. 3). In quartets that were most closely matched in dominance rank, those with the highest *Group Social Tolerance* were estimated to sustain the resource for a mean of 97 s longer than those with the lowest *Group Social Tolerance* (95% CI [11.2, 179], pd % = 98). A full posterior predictive contrast incorporating outcome-level variance found an average difference of 97 s (95% CI [11.5, 179], pd% = 98). In groups with the median level of *Group Dominance Difference*, the contrast in *Last Stick Latency* predicted by *Group Social Tolerance* moving from lowest to highest was smaller and less strongly supported, with a mean difference of 72 s (95% CI [−1.18, 138], pd% = 97). A full posterior predictive contrast incorporating outcome-level variance found an average difference of 71 s (95% CI [−1.7, 138], pd% = 97). In quartets with the highest degree of dominance asymmetry, there was no effect, with an estimated mean 3 s contrast between the lowest and highest *Group Social Tolerance* (95% CI [−115.9, 219.0], pd% = 51). Therefore, if individuals within a group were closely matched in dominance rank, their level of social tolerance predicted their success at the task (the time until the common-pool resource would be collapsed). If there was a high degree of dominance asymmetry in the group, tolerance did not predict success.

In the dyads, latency was not strongly predicted by dominance (β = −0.17, 95% CI [−0.50, 0.18], pd% = 84) or tolerance (β = −0.15, 95% CI [−0.46, 0.16], pd% = 84) or their interaction (β = −0.10, 95% CI [−0.50, 0.32], pd% = 68).

**Dominance and payoff interaction.** To investigate the possible mechanisms underlying this dominance-tolerance interaction effect in quartets, we conducted a post-hoc analysis (Model 4). The central question of this analysis was whether lower dominance-difference quartets benefit from higher social tolerance, but high-dominance-difference quartets do not. Are the groups flexibly achieving success through different strategies depending on the social dynamics of the group?

We hypothesised that, in high *Group Dominance Difference* groups, success is achieved through monopolisation by the dominant individual, as reported by Koomen and Hermann[22]. Therefore, in this dominance-

tolerance configuration, latency until resource collapse will be positively associated with the relative payoff of the highest-ranked individual in the group. In other words, dominants secure their own access to the pool and are able to exclude more subordinate players in the process, and, at the group level, this avoids over-extracting from the common-pool resource, extending latency until collapse.

We anticipated a different mechanism in low-dominance-difference-high-tolerance groups, where dominant apes have less social leverage and may be less willing or able to exclude other group members from accessing the pool. We hypothesised that the underlying explanation for *Group Social Tolerance* supporting success in low *Group Dominance Difference* groups is that dominant individuals are more likely to accept low payoffs in these groups. Therefore, latency until collapse would be negatively associated with the relative payoff of the highest-ranker.

The proportion of time that the dominant individual held sticks relative to the other participants was calculated. Duration of holding time was used in place of actual count of scoops acquired, as this measure accounted for individual differences in speed and dexterity that may have affected count scores, though the two measures were highly correlated ($r_s = 0.96$, 95% CI [0.94, 0.97], $p = <0.001$, $n = 103$). Model 4 was then constructed, examining the three-way interaction between proportion (of stick-holding by the highest-ranked individual), within-group dominance-difference, and within-group social tolerance.

This hypothesis was not supported. In quartets, it was found that overall (not only in low-dominance-difference-high-tolerance groups), but trials that were sustained for longer were also associated with the dominant individual extracting less from the resource relative to the other players. When the *Proportion* (of stick holding attributed to the dominant individual) was at its lowest, the resource was sustained on average 80 s longer (95% CI [−164.0, 14.7], pd% = 96) than when the *Proportion* was at its highest. A full posterior predictive contrast incorporating outcome-level variance found an average difference of −80 s (95% CI [−164, 14.1], pd% = 96). There was no strong evidence that the interaction between *Proportion* and *Group Social Tolerance* (β = 0.06, 95% CI [−0.28, 0.18], pd% = 70) or *Group Dominance* (β = −0.17, 95% CI [−0.43, 0.08], pd% = 90) affected *Collapse Latency*.

Model 5 applied this analysis to the dyad dataset and found no clear effect of *Proportion* consumed by the dominant individual on *Collapse Latency* (β = 0.04, 95% CI [−0.05, 0.12], pd% = 79).

**Payoff equality**. Payoffs were measured by counts of scoops of yoghurt consumed. Our pre-registered hypothesis (H4) stated that payoffs would be less equal in dyads compared to quartets, especially in dyads with high *Group Dominance Difference* and low *Group Social Tolerance*.

The GLMM (Model 6) found group size to be positively associated with payoff equality, meaning that, as hypothesised, when accounting for differences in *Collapse Latency*, payoffs were less equal in dyads compared to quartets by an average of 49% (95% CI [21, 81], pd% = 100) (0.22 change in the entropy unit). A full posterior predictive contrast incorporating outcome-level variance found an average difference of 0.18 (95% CI [0.05, 0.29], pd% = 100). Trials that were sustained for longer tended to have lower payoff equality (β = −0.71, 95% CI [−0.88, −0.54], pd% = 100), with a stronger effect in dyads. Between the lowest and highest *Collapse Latency* trials, payoff equality (*Normalised Entropy*) was estimated to decrease by 84% in dyads (95% CI [−88, −79], pd% = 100), but only 13% in quartets (95% CI [−38, 20], pd% = 81). The interpretation of this is straightforward; as the participants did not share sticks or establish turn-taking, long dyadic trials will intrinsically be unequal, resulting in this strong interaction. If one participant is feeding, in order for the second player to feed, they must remove the second stick, triggering resource collapse. Therefore, in most instances, the longer the trial runs for, the more unequal it will become, creating a strong direct relationship between latency and equality. A full posterior predictive contrast incorporating outcome-level variance found the difference in *Normalised Entropy* between highest and lowest *Collapse*

*Latency* to be −83% (95% CI [−90, −77], pd% = 100) in dyads and −13% (95% CI [−40, 24], pd% = 78) in quartets.

The interacting effect of *Group Dominance Difference* and *Group Social Tolerance* was tested on subgroups based on group size (Models 7 and 8), with no clear effects. A possible negative effect of tolerance in quartets was identified, with payoff equality reducing an estimate of 12% (95% CI [−24, 7], pd% = 92) between the least tolerant and most tolerant groups. A full posterior predictive contrast incorporating outcome-level variance found this difference to be −11.3% (95% CI [−24.5, 7], pd% = 91).

## Discussion

We find evidence that chimpanzees adjust their behaviour to promote group-level resource sustainability in common-pool resource dilemmas. However, like humans, the level of success that the chimpanzees achieve, and the most successful strategies to engage, can vary greatly with the social dynamics of the group and the structure of the common-pool resource dilemma itself.

Quartets of chimpanzees took on average 101% longer to remove all sticks from the common-pool resource in the test condition, compared to the control. This indicates their ability to sustain the resource in the social dilemma test condition, despite a tendency to remove the sticks more quickly in the non-social-dilemma condition. Grouped as dyads, the same apes were not as successful at delaying resource collapse, both in terms of overall trial latency and in the difference in latency until all sticks were removed in the two conditions (28.5% longer in the test compared to the control). As the 95% credible interval for the effect of condition on latency until the last stick removed does not entirely exclude zero in the dyad model, this could indicate that the apes do not distinguish between the control condition and the social dilemma. However, the clear difference between conditions when the same chimpanzees were grouped as quartets, in addition to comprehension criteria that they fulfilled prior to testing (detailed in SI), makes this interpretation implausible.

If the sticks were removed at random intervals, independently of one another, it would be expected that collapse would happen later if four sticks were present, compared to two. This is likely reflected in the difference between the control conditions: all sticks are removed, on average, 38.2 s later in quartets compared to dyads. However, the difference in this measure between group sizes in the test condition is much greater (109 s), suggesting that this is unlikely to fully explain the group size difference.

The difference between dyads and quartets may be partially attributed to space constraints. The apparatus occupied the same space in the testing room in both group-size conditions, requiring the quartets to feed in closer proximity. This perhaps gave the quartets an advantage, as it presented more of a cofeeding challenge, more opportunity to exclude other players, or just greater salience that players' fates are intertwined. Furthermore, success in this task can be achieved by just one participant declining to remove a stick, and some participants are more likely to fill this role than others. Quartets may therefore have an advantage over dyads simply because having more players increases the chances of having one of these players in the group. However, this would not explain why the dyads containing the particularly able individuals fail to avoid collapse as long as the best quartets.

One possibility is that chimpanzees are less likely to inhibit competitive tendencies when grouped in dyads. In a review of work on the evolution of "group-mindedness", Brooks and Yamamoto[25] reflect on the discrepancy in findings between dyadic experimental work on cooperation, that finds chimpanzees unwilling to engage in even non-costly reciprocity in prosocial choice tasks[42–44], and the impressive cooperative feats that wild groups of chimpanzees are capable of, including those that require the resolution of social dilemmas, such as group hunting and territory defence[45,46]. Brooks and Yamamoto[25] suggest that chimpanzees have more competitive tendencies in the dyad; an adaptation to the unpredictable foraging environment they have evolved in[47]. Their group-level cooperative behaviours may have been shaped, not by building on dyadic cooperative tendencies, as is proposed for humans[48], but by a need to resolve group-level Collective Action Problems, such as territory defence. According to this perspective, by

placing chimpanzees in dyads, experimenters may actually curtail the cooperative abilities of chimpanzees[25,28,49].

Our findings generally agree with this perspective. In quartets, but not dyads, higher social tolerance is predictive of longer collapse latencies. Additionally, longer quartet, but not dyad, trials are associated with the dominant apes achieving lower payoffs. This suggests that the dominant chimpanzees are able to employ tolerant and cooperative strategies in the quartet, but are unwilling or unable to inhibit competitive behaviour in the dyadic experimental setup. The current experiment was not designed to test this theory, termed "top-down group cooperation" by Brooks and Yamamoto[25]; however, our findings indicate a promising area warranting further research.

The finding that quartets with higher social tolerance are more able to resolve the social dilemma than those with low social tolerance is consistent with recent findings from Leeuwen et al.[50] that sanctuary-housed chimpanzees engage in more prosocial behaviour in a group-service paradigm in groups with a higher level of social tolerance. This task differs from our paradigm in the sense that it requires an individual to generate a benefit for the group, while the common-pool resource task requires an individual to resist an ultimately negative outcome for the group. However, the two experiments include a similar mechanism, where group benefits are interdependent on individual decisions. Interestingly, and in further support of top-down group cooperation, this group-level study found chimpanzees to be more prosocial than equivalent assigned-partner dyadic studies[43]. For wild chimpanzees, too, resolving the naturalistic collective action problem of territory defence is supported by in-group cohesion, which includes tolerance[51,52].

Tolerance was found to be especially supportive of common-pool resource dilemma success in quartets where participants were closer in rank in the dominance hierarchy. Quartets of apes that were close in dominance rank but had low social tolerance were relatively unsuccessful at the task. By contrast, quartets with greater differences in dominance rank were found to be successful, even in the absence of social tolerance. Our hypothesis (hypothesis 3) that high dominance differences and low social tolerance would promote resource sustainability was therefore only partially supported. Our pre-registered hypotheses rested on an assumption, informed by past *dyadic* work[22], that success through monopolisation by the most dominant individual would be the best strategy for the chimpanzees; however, our findings produce a more complicated image of chimpanzee resource management. Even in the successful quartets with low tolerance and high dominance difference, we find no evidence that success was achieved through monopolisation by the dominant ape, with lower rankers unwilling to extract from the resource in their presence due to low social tolerance. Rather, longer resource sustainability was negatively associated with the payoff of the dominant individual, independent of social tolerance and dominance difference. Therefore, while the dominant's behaviour *was* predictive of group success, the tolerant act of allowing others to feed for longer than them appears to be the key to sustainable outcomes. For quartets, one of the strongest predictors of success was having a high-ranking individual in the group who was not strongly motivated to monopolise the resource. There is modest evidence that unequal payoffs *were* associated with higher latency among the quartets; however, it seems that the dominant individuals tended not to be the main beneficiaries in these higher-latency-higher-payoff-inequity trials.

Among humans, equitable payoffs are consistently associated with common-pool resource longevity[1]. The failure of chimpanzees to find mutually beneficial strategies, despite social tolerance, is indicative of a difference between human cooperation and chimpanzee cooperation. Equity is of such importance to human commons management that theorists will sometimes include it in their definition of "success", for example, when non-renewable resources, such as coal deposits, are framed as common-pool resource dilemmas, emphasis is placed on intergenerational equity rather than renewability[53]. Our findings concur with prior evidence that chimpanzees do not share humans' sense of "fairness" as the resource was best sustained in trials with more unequal payoffs[54].

## Limitations

Findings on chimpanzee prosociality are notoriously inconsistent[55,56]. Recent findings on intergroup differences in tolerance and how it relates to measurable acts of prosociality suggest that these inconsistent findings arise from single-group testing[57,58] Human behaviour around common-pool resources is culturally mediated[59] and, based on intergroup chimpanzee prosociality findings, it is likely that the way chimpanzees respond to common-pool resources also varies flexibly across groups. Expanding testing to additional populations may improve the robustness of our findings.

There are also limitations associated with how the social variables were measured. It was not possible to run the same dominance test used by Koomen and Herrmann[21], so a keeper survey was used instead. We acknowledge that, although the four keepers surveyed had extensive experience with the apes, this is a less objective measure. Similarly, our tolerance measure was based on proximity data to quantify association, while in Koomen and Herrmann[22] it was calculated from a cofeeding tolerance test. While this methodological difference may preclude strict comparison of the findings, we believe both approaches capture the quality of dyadic relationships in a way that is distinct from hierarchical differences and generalisable across paradigms.

Finally, to calculate *Group Dominance Difference* and *Group Social Tolerance* for groups of four, the mean dyadic dominance-difference and dyadic association index of the six dyads that made up the quartet were taken. Information about variance is lost when these values are collapsed into a mean.

## Conclusion

In the first experimental study of groups (>2) of chimpanzees in a common-pool resource dilemma, and one of the few group-level comparative behavioural economics studies, our expectation that the performance of groups would hinge on group size, tolerance, and dominance was correct. However, the effect of these social variables on group sustainable behaviour manifested in unexpected ways.

Generally, chimpanzee quartets with higher social tolerance were better at resolving the dilemma. The importance of tolerance was somewhat reduced when the degree of dominance difference was increased, and higher-dominance-difference groups were also found to achieve high collapse latencies, even when tolerance was low. This would perhaps imply two different strategies that are flexibly adopted: in high dominance-difference groups, success through monopolisation by the dominant, and in low dominance-difference groups, success through non-competitive cofeeding tolerance. However, when this theory was tested, we instead found that dominant apes achieving low payoffs relative to the other players was predictive of high latency, regardless of group tolerance and dominance relationships.

This effect did not hold when the apes were grouped as dyads. Furthermore, there was only modest evidence that dyads behaved differently in the common-pool resource dilemma than if they were extracting resources in the absence of a social dilemma, as inferred by no big difference between latencies in the test condition compared to the control. Overall, chimpanzees were less able to prevent resource collapse as dyads in comparison to when they were grouped as quartets.

Further testing of the paradigm with human children is already underway (https://osf.io/agczu/). Comparison with the existing human literature indicates that the strategies of successful chimpanzee groups differ from those of humans. Consistent with prior work, the chimpanzees did not develop turn-taking strategies[60], and sustainable resource use was associated with unequal payoffs. Koomen and Herrmann[22] have compared this to how humans may privatise resources to prevent over-use, a management strategy that usually involves excluding those with less power and money. However, our study finds that the low payoffs may be voluntarily accepted by the chimpanzee with the *most* leverage, the dominant one. Although the underlying motivations for this behaviour remain unclear, this finding highlights that, while human cooperation is uniquely sophisticated,

**Article**

chimpanzees can also resolve complex social dilemmas, employing a different set of strategies.

## Data availability
All data used in the analysis is available at https://doi.org/10.17605/OSF.IO/Q8GA6[61].

## Code availability
R code is available at https://doi.org/10.17605/OSF.IO/Q8GA6[61].

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

## Acknowledgements
This research was funded by the Max Planck Society for the Advancement of Science. The funders had no role in study design, data collection and analysis, decision to publish or preparation of the manuscript. We would like to thank the animal carers and scientific staff at the Wolfgang Köhler Primate Research Centre at Leipzig Zoo, with a special mention to Raik Pieszek, for building the apparatus. We thank Jade Czesnick for reliability coding, and Keti Robakidze and Nikita Raphael Kern for assistance in preparing video material. Milana Gries produced Fig. 1. We also thank Luke Maurits for statistical support and designing the power analysis.

## Author contributions
K.S.: conceptualisation, data collection, analysis, project administration, writing – original draft, writing – editing. D.H.: conceptualisation, funding acquisition, supervision, writing – review & editing. A.S.: conceptualisation, funding acquisition, supervision, writing – review & editing.

## Funding

## Competing interests
The authors declare no competing interests.
