## [Transparent Peer Review file · Communications Psychology]

Chimpanzee groups achieve sustainable resource use in a Common Pool Resource dilemma

Corresponding Author: Ms Kirsten Sutherland

Version 0:

Decision Letter:

Dear Ms Sutherland,

Thank you for your patience during the peer-review process. Your manuscript titled "Tolerance in chimpanzee groups leads to more sustainable resource use in a Common Pool Resource dilemma" has now been seen by 3 reviewers, and I include their comments at the end of this message. They find your work of interest but raised some important points. We are interested in the possibility of publishing your study in Communications Psychology, but would like to consider your responses to these concerns and assess a revised manuscript before we make a final decision on publication.

We therefore invite you to revise and resubmit your manuscript, along with a point-by-point response to the reviewers. Please highlight all changes in the manuscript text file.

Editorially, we consider it crucial that the hypotheses and the methodological details are made clearer in the revised manuscript. Please also address reviewer #3's concerns regarding how the individual-level correlation is not accounted for in the statistical analysis.

I am attaching an Editorial Requests Table that details critical reporting requirements for the revised manuscript. Please attend to each item and ensure your manuscript is fully compliant. If your revised manuscript is not aligned with these requests on major issues, such as those concerning statistics, it may be returned to you for further revisions without re-review.

Please submit the following items:

- Revised manuscript
- Point-by-point response to the referees' comments
- Cover letter (as a separate document)
- <https://www.nature.com/documents/nr-reporting-summary.pdf> Nature Research Reporting Summary
- Completed Editorial Request Table (attached).

via this link: Link Redacted .

Additional guidance is available in our style and formatting guide Communications Psychology formatting guide.

Best regards,

Troy Lui

Troy Lui, PhD
Associate Editor
Communications Psychology

REVIEWER EXPERTISE:

Reviewer #1: primate, social cooperation

Reviewer #2: primate, social cooperation

Reviewer #3: statistics

REVIEWER REPORTS:

Reviewer #1 (Remarks to the Author):

This is an interesting, well-written paper that is also fairly complicated and somewhat confusing. Most of the confusion revolves around the presentation of the methods employed to address the multiple hypotheses in the study. The results are also somewhat confusing. In general, the authors were interested in exploring a number of variables/hypotheses related to the Common Resource Pool (CRP) dilemma in nonhuman primates. In this study, captive chimpanzees living in Leipzig, Germany were the subjects. The authors devised an ingenious apparatus that allowed them to assess how various social variables might affect chimpanzees feeding from a device that would 'collapse' if all of the animals attempted to feed from it. One of the major behavioral comparisons was between the responses of chimpanzee dyads (n=17) vs. quartets (n=7) drawn from two separate social groups. A large number of trials (708 over a 15-month period) were performed on a reasonable number of individuals (10 in one group and 5 in the other).

The manuscript is approximately 28 pages long and is accompanied by approximately 25 pages of Supplementary information. It was necessary to read most of the Supplementary information to understand the way in which the study was conducted (the procedures that were employed). This suggests that significant portions of the Supplementary information probably belonged in the body of the manuscript itself. This would have made even an on-line version of the manuscript quite lengthy. The study was pre-registered, and the authors explain effectively, and in considerable detail, their deviations from the pre-registration.

Unfortunately, this reviewer has limited expertise in the statistical techniques that were employed to analyze the data. Therefore, it is difficult for this reviewer to accurately assess the value of the findings. While most of the results make sense, this reviewer cannot effectively comment on whether the data support or refute the multiple hypotheses proposed and addressed in the study. This is obviously a shortcoming of the reviewer. Hopefully, another reviewer can address the statistical approach.

However, this reviewer is qualified to comment on the provided descriptions of the methods of the study (see below).

Comments on the writing:

There are a few passages where verb tenses shift back and forth in the same sentence/paragraph.

There are a fair number of typos in the supplementary materials.

Potentially relevant questions/comments:

Group B seemed to require considerably fewer P-E sessions until pass to 'get it'. Do the authors have any thoughts on why that might be?

Lines 122-125 It is not entirely clear that this CPR is a simple and intuitive paradigm.

Line 138 When did the trial actually start; when the door opened, allowing the chimpanzees into the test room or when the first chimpanzee touched the first stick? According to

Lines 425-426 and mentioned in several other spots, the trial started when the door between the rooms opened. In Lines 602-604 – is it correct that lid latency began when the trial began (when the door opened)? Might it have made more sense to have lid latency begin when the first stick was touched? Did chimpanzees IMMEDIATELY head for and touch a stick when the door to the test chamber was opened? If so, it might be worthwhile to state this.

Lines 139-141 How did the chimpanzees know it was a control trial? Were the sticks positioned differently than during a test trial? Or was it simply that in the control condition, the lid was not present?

Interobserver reliabilities were really, really high; much higher than one typically sees in behavioral research.

Figure 3 is fairly difficult to interpret.

Might straws have been better than sticks? It would not have been necessary to remove straws, which eliminates the collapse aspect of the CPR, so probably not. But could the animals re-insert a stick, using it to stop the lid from coming down and closing off the resource during the approximately 10 seconds it took for the lid to close?

The authors state on lines 471-472 that they did not observe the chimpanzees adopting turn-taking strategies. Do they have any thoughts about why this potentially quite successful strategy was not adopted? Similarly, if animal A in a quartet did not remove a stick in one trial, did animal B 'sacrifice' itself in the next trial of the same quartet and not remove a stick, allowing animal A to eat? Then animal C, etc.?

Some of the reporting of the results (Lines 307-314), especially of 3-way interactions, gets confusing.

Line 348 What does it mean to be "good" at the task? Not removing a stick, so that the others can eat?

Lines 529-535 A session contained 1-5 trials. How were trials separated from one another? Were the chimpanzees moved out of the testing room and put back in the waiting room? Did each trial begin with a full allocation of yoghurt?

Do the authors find it surprising that there was no learning effect across sessions?

The pre-experience photo is a little confusing. How many sticks are there in this photo?

Reviewer #2 (Remarks to the Author):

Tolerance in chimpanzee groups leads to more sustainable resource use in a Common Pool Resource dilemma

General Comments

This paper was genuinely a delight to read. The apparatus design is intriguing, the language of the manuscript is relatively jargon-free and accessible to non-specialists, and the group size comparison indeed adds significantly to our understanding of the CPR behavioural characteristics of chimpanzees. Minor spelling/grammar errors throughout. Minor to moderate revision suggestions are provided for each section, below. No major revision suggestions are provided. The contributions of this paper are significant, particularly in light of the findings of Koomen & Herrmann, 2018b. Namely, the following contributions are noteworthy and important for the field of comparative psychology: the clever apparatus design, the dyadic vs quartet comparison, and the findings on a dominance difference * tolerance interaction in quartets. I applaud the authors for the work.

Introduction

P4 Line 108-109 – three studies were identified but then only two are cited?

P4 line 123 – collapse mechanism of study 1 is not arbitrary, it is linked directly and visually with the amount of the resource that is available at any given time

P6 line 152 – citation should be 2018b, not c

P6 line 167-169 – Hypothesis 2: clarify which outcomes are being referred to in relation to which interpretation.

P7 – Hypothesis 4: it's unclear to me why payoffs are predicted to be more unequal in dyads than in quartets

Hypothesis 2 is generally unclear to me. Why would apes approach and extract more quickly from the control than the CPR dilemma? Couldn't the opposite prediction also follow logically from the scramble competition involved in the CPR dilemma? Also, why is approach not actually measured, but only total consumption time (ie latency to last stick)? Wouldn't latency to first stick be a more direct measure of approach while also being more comparable across conditions? Latency to last stick in the control condition is constrained only by the groups' feeding preferences while latency to last stick in the CPR condition is constrained by the groups' feeding preferences interacting with the physical affordances of the dilemma. I'm not sure how this is a measure of the chimpanzees' awareness of the dilemma.

This is a matter of personal opinion, but I no longer cite Hardin because of his white supremacist writings. One way I have found to discuss the 'tragedy' concept is to cite the work of others who cited him (... As cited in X).

Results

Quartets sustained the CPR for longer latencies than the dyads – this is what one would predict from random use of the resource because with four sticks in comparison to two, one would expect that collapse occurs less quickly when all sticks (2 vs 4) are taken at random intervals independently of one another.

Figure 3 and the ad-hoc analysis looking at relative payoff of the dominant individual provide a fascinating look at the dominance:tolerance interaction in chimpanzee CPR behaviour; certainly sheds new light on the findings of Koomen & Herrmann, 2018 (chimpanzee CPR paper).

Discussion

P13 – “by placing chimpanzees in dyads, experimenters may actually curtail the cooperative abilities of chimpanzees..” – this is an important insight from this study: The finding that sustainability success differences are found between dyads and quartets (and the novel comparative apparatus) provide a significant step forward in our understanding of the CPR behaviours of our closest living relatives.

P14 line 384-385 – ‘facilitates’ suggests an active role in both paradigms. Suggested: “However, the two experiments involve a similar mechanism, where group benefits are interdependent upon individual decisions.”

P16 penultimate paragraph – I would advise the authors against speculating on the lessons to be learned from a data set that does not yet exist.

Methods

Dominance is measured in this study via keeper reports of relative rank. Given that hypotheses are based on findings from Koomen & Herrmann, 2018b, it should be stated clearly that the quantitative measure of dominance difference in this study, and the binary rank differences in the dyadic study (Koomen & Herrmann, 2018b) differ.

Same comment for the measurement of group-level / dyad-level tolerance. In the present study this is a behavioural measure based on physical proximity while the tolerance measure in Koomen & Herrmann's dyads was a co-feeding measure based on duration of co-feeding from a shared pile of food. If direct comparisons are made between the findings of the three studies, it should be clearly stated that the social measures may be capturing a different quantitative impression of the relationships being compared. Perhaps the authors can also justify why they opted for these measures?

Were gaussian distribution GLMMs compared for fit against the gamma distribution family for the latency outcome data? Some CPR outcome data from human studies are more closely fit with gaussian distributions.

Supplementary Information

Looks very well organised and detailed.

Additional comment:

Given the nebulous nature of tolerance and the many ways it is operationalised in non-human primates (e.g., DeTroy, Haun, & van Leeuwen, 2022), given the fact that in this paper, tolerance is operationalised as time spent in proximity and it is being compared directly with findings from Koomen & Herrmann (2018), which operationalised co-feeding tolerance as time spent in proximity while co-feeding on a shared food source, AND given that the tolerance measure in the present study forms a central part of the main conclusions, I would caution the authors against making this claim in the title itself without a bit of clarification. Is, for example, the long-term Leipzig data on proximity more comparable with Engelmann & Herrmann (2016) operationalisation of "friendship" (SCI)? I'm not sure there is a straightforward answer here, because the measurement of

relationship quality in non-human primates, as in humans, is a complex and sometimes fraught endeavour, but the title as it currently stands doesn't seem appropriate and the discussion on the use of 'tolerance' in the paper is insufficient. This does not, however, take away from the importance of the findings.

Reviewer #3 (Remarks to the Author):

Please see the "...AuthorComments.pdf" file attached.

Version 1:

Decision Letter:

Dear Ms Sutherland,

Your manuscript titled "Chimpanzee groups achieve sustainable resource use in a Common Pool Resource dilemma" has now been seen by our reviewers, whose comments appear below. In light of their advice I am delighted to say that we are happy, in principle, to publish a suitably revised version in Communications Psychology.

We therefore invite you to revise your paper one last time to address the remaining concerns of our reviewers and a list of editorial requests. At the same time we ask that you edit your manuscript to comply with our format requirements and to maximise the accessibility and therefore the impact of your work.

EDITORIAL REQUESTS:

SUBMISSION INFORMATION:

OPEN ACCESS:

Communications Psychology is a fully open access journal. Articles are made freely accessible on publication. For further

information about article processing charges, open access funding, and advice and support from Nature Research, please visit <https://www.nature.com/commpsychol/open-access>

* **DATA AVAILABILITY:**

All Communications Psychology manuscripts must include a section titled "Data Availability" at the end of the Methods section. More information on this policy, is available in the Editorial Requests Table and at <http://www.nature.com/authors/policies/data/data-availability-statements-data-citations.pdf>

Link Redacted

Best regards,

Troby Lui

Troby Lui, PhD
Associate Editor
Communications Psychology

REVIEWERS' COMMENTS:

Reviewer #1 (Remarks to the Author):

This is a revision of an interesting, well-written paper that explored a number of variables/hypotheses related to the Common Pool Resource (CPR) dilemma in nonhuman primates. In this study, captive chimpanzees living in Leipzig, Germany were the subjects. The authors devised an ingenious apparatus that allowed them to assess how various social variables might affect chimpanzees feeding from a device that, during the test condition, would 'collapse' if all of the animals attempted to use the sticks propping up the lid to feed from it. During the control condition, there was no lid, so the animals could remove and use the sticks with no adverse consequences. One of the major behavioral comparisons was between the responses of chimpanzee dyads (n=17) and quartets (n=7) drawn from two separate social groups. A large number of trials (708 over a 15-month period) were performed on a reasonable number of individuals (10 in one group and 5 in the other). The authors have successfully addressed most of the comments from the original submission. A small number of questions still remain, primarily for clarification purposes.

The manuscript is still quite long with a considerable amount of important supplementary information, especially concerning the experimental methods.

Did the chimpanzees ever show aggressive/frustration responses once the lid had closed? The mesh barriers prevented them from 'attacking' the lid, but did they try to get more if yogurt was still present?

Figure 1 is good.

How were non-study chimpanzees excluded from the 'waiting' room? The waiting room was not the same as the 'colony' room, correct? If a chimpanzee chose not to participate, how could s/he return to the rest of the social group?

What level of agreement existed among the four keepers that evaluated the animals' relative dominance?

Do the authors have any data on the order in which chimpanzees used the sticks? Did the 'dominants' routinely have the lowest latencies to stick use?

What proportion of time was at least one chimpanzee 'eating' during the period prior to resource collapse?

Lines 233-237 The descriptions of the dependent variables in this study are confusing. From entry to lid collapse is the dependent measure for experimental trials, correct (Model 1)? Is entry to removal of the last stick (from the pool) the dependent measure for control trials? This should be clarified. If the dependent variable is the removal of the last stick from the pool, are these two dependent measures likely to be comparable? It seems that Models 2 and 3 used removal of the last stick as the dependent measure?

Lines 490-491 Might it be that dominant individuals are more likely to tolerate higher payoffs in the other members of the group?

SI Line 109 Do the authors have any thoughts concerning why performance did not improve over sessions (a learning effect)?

SI Lines 272-274 Wouldn't recording these latencies as 10 minutes make it seem as though the animals 'succeeded' at cooperating?

Reviewer #2 (Remarks to the Author):

Reviewer 2 Response to Authors.

The authors have adequately addressed my concerns with the exception of the 4 points below. Once these are sufficiently addressed, I will recommend acceptance. I extend gratitude to the authors, in advance, for their contribution to the CPR literature.

1.

Ms Line 3423. "does not require participants to undergo extensive training in order to participate (such as the collapse rule in Study 1 of Koomen and Herrmann, 2018b)."

Reviewer response: The implication here is that the participants were 'trained' to make sustainable choices which is not accurate, but this also precludes the possibility that the task demands rendered the DoG more challenging (i.e. stop sucking on delicious juice may be harder behaviourally than the choice to avoid picking up a stick to get delicious yoghurt). I would suggest: "to test this experimentally with simpler task demands than study 1 of koomen & herrmann" because in both paradigms it is impossible to tease apart the effect of DoG difficulties from comprehension difficulties and the use of 'task demands' remains conservatively ambivalent about this distinction.

2.

Ms Line 3595. "While our focus on dominance and tolerance dynamics draws from Koomen and Herrmann (2018b), we use different measures to investigate them. Koomen and Herrmann (2018b, Study 2) conducted a pre-test where their dyads of chimpanzees were given simultaneous access to a pile of carrot pieces. The individual that was able to consume more pieces was rated as the dominant, and the co-feeding tolerance measure was based on the latency of the subordinate individual to feed at the closest possible location to the dominant. It was not possible to conduct this type of pre-test at Leipzig Zoo, due to concerns that it could elicit aggression."

Reviewer response: The framing of this paragraph might lead readers to conclude that unethical practices were conducted. I would urge the authors to simplify this paragraph by reducing the description of the methods differences to 'co-feeding tolerance' (and cite some of the many previous studies who have operationalised tolerance this way) and 'proximity' tolerance (or some other term that differentiates it from co-feeding). And the authors can then say that due to institutional differences between the ape facilities, co-feeding tolerance measures were not possible.

Additional Reviewer response: Line 4712 -> "similarly, our tolerance measure was proximity-based and not co-feeding based as it was in koomen & herrmann. While this methodological difference may preclude strict comparison of the findings, we believe both tolerance methods measure something important about the quality of dyadic relationships that is distinct from hierarchical differences and generalisable across paradigms."

3.

Reviewer comment. Pages 10 – 11 of the authors' Response to Reviewers states the below, but I don't see any of this clarification reflected in the manuscript itself. This needs to be clarified in the ms.

6.2 "Couldn't the opposite prediction also follow logically from the scramble competition involved in the CPR dilemma?"

This comment may be resolved already by removing the word "approach" from Hypothesis 2, however we have elaborated specifically on the reviewer's question relating to scramble competition:

Based on previous findings (Koomen & Herrmann 2018b) we, firstly, expected some groups of apes to successfully

overcome the social dilemma and delay resource collapse. We did not expect them to develop turn-taking or stick-sharing strategies as humans might. Therefore, we expected that the most likely solution for them would be to leave one stick inside the pool to keep it open. Secondly, we expected more dominance-asymmetric and lower tolerance groups to leave one stick inside for longer as the more dominant apes monopolised the resource (also based on Koomen & Herrmann 2018b). Therefore, the feeding competition that emerges in the test condition is more closely modelled by contest competition than scramble competition, though, like most real-world scenarios, it's really a combination of both (van Schaik & Noordwijk, 1988).

Scramble feeding may be a more accurate model for the control condition, as a motivated ape should "scramble" to take a stick as quickly as they can, before another participant claims it.

4.
Reviewer comment. Page 12 of the authors' Response to Reviewers states the below, but again from the authors' response it appears as though no clarification was added to the manuscript itself. This needs to be clarified in the ms.

6.4 "Latency to last stick in the control condition is constrained only by the groups' feeding preferences while latency to last stick in the CPR condition is constrained by the groups' feeding preferences interacting with the physical affordances of the dilemma. I'm not sure how this is a measure of the chimpanzees' awareness of the dilemma."

It is not intended as a measure of their awareness of the dilemma, but as a measure of their ability to sustain the resource in the test condition, in comparison to their preference in the absence of a social dilemma in the control. The difference in latency until last stick removal illustrates that the participants do not purely scramble to get the sticks as fast as possible, but modify their behaviour in response to the social dilemma.

Reviewer #3 (Remarks to the Author):

I am satisfied with the author's revisions based on the original comments I had made. My main concern was with the random effect for individual, and since that was addressed appropriately, I think the statistical methods used are now suitable. I appreciate the response and updates to my additional comments as well.

Dear Editors of Communication Psychology,
Dear Dr. Lui,

Thank you very much for giving us the opportunity to submit a revised version of our manuscript titled *Chimpanzee groups achieve sustainable resource use in a Common Pool Resource dilemma* (previously, *Tolerance in chimpanzee groups leads to more sustainable resource use in a Common Pool Resource dilemma*). We were very grateful for the valuable and constructive feedback of the reviewers, which we feel has significantly improved our manuscript. We greatly appreciate that the reviewers saw considerable merit in our work.

In the revised version of the manuscript, we address all concerns raised by the three reviewers. In the following we outline our revisions point by point. Our revisions in this response letter are highlighted in *blue*.

Reviewer 1

(1)

“This is an interesting, well-written paper that is also fairly complicated and somewhat confusing. Most of the confusion revolves around the presentation of the methods employed to address the multiple hypotheses in the study. The results are also somewhat confusing. In general, the authors were interested in exploring a number of variables/hypotheses related to the Common Resource Pool (CRP) dilemma in nonhuman primates. In this study, captive chimpanzees living in Leipzig, Germany were the subjects. The authors devised an ingenious apparatus that allowed them to assess how various social variables might affect chimpanzees feeding from a device that would ‘collapse’ if all of the animals attempted to feed from it. One of the major behavioral comparisons was between the responses of chimpanzee dyads (n=17) vs. quartets (n=7) drawn from two separate social groups. A large number of trials (708 over a 15-month period) were performed on a reasonable number of individuals (10 in one group and 5 in the other).

The manuscript is approximately 28 pages long and is accompanied by approximately 25 pages of Supplementary information. It was necessary to read most of the Supplementary information to understand the way in which the study was conducted (the procedures that were employed). This suggests that significant portions of the Supplementary information probably belonged in the body of the manuscript itself. This would have made even an on-line version of the manuscript quite lengthy. The study was pre-registered, and the authors explain effectively, and in considerable detail, their deviations from the pre-registration.”

Unfortunately, this reviewer has limited expertise in the statistical techniques that were employed to analyze the data. Therefore, it is difficult for this reviewer to accurately assess the value of the findings. While most of the results make sense, this reviewer cannot effectively comment on whether the data support or refute the multiple hypotheses proposed and addressed in the study. This is obviously a shortcoming of the reviewer. Hopefully, another reviewer can address the statistical approach.”

We thank the reviewer for the positive comments about the paper, and are pleased to read that they found the apparatus “ingenious”.

We have made improvements to the presentation of the methods based on the reviewer’s comments (see points 6, 13 and 15).

We agree that some of the results are not simple. We have made changes to the structure of the results reporting to improve clarity (see point 13). We now present findings with a description of the effect, followed by model estimates. We have not pasted all the changes in the response letter because they were mainly restructuring rather than completely new text, and they can be seen throughout the results section.

(2)

“There are a few passages where verb tenses shift back and forth in the same sentence/paragraph.”

We have re-read the entire paper and changed all reporting to past tense (except for conditional tense discussion of hypotheses and hypothetical outcomes).

(3)

“There are a fair number of typos in the supplementary materials.”

Thank you for pointing this out. We have more carefully proof-read the SI and corrected typos.

(4)

“Group B seemed to require considerably fewer P-E sessions until pass to ‘get it’. Do the authors have any thoughts on why that might be?”

We expect that this is simply an artefact of a small number of chimpanzees in the B-group. The modal number of sessions until passing is the same for A-group and B-group (3 sessions).

(5)

“Lines 122-125 It is not entirely clear that this CPR is a simple and intuitive paradigm.”

In this section we describe the aims of the study, one of which was to design an apparatus that was intuitive for the chimpanzees in order to attain as large a sample size as possible.

We agree with the reviewer that whether we succeeded is a matter of opinion, however, we would argue that, as stated in line 142, relative to the collapse mechanism of Study 1 in

Koomen and Herrmann (2018b), our design is simpler for the participants, as the apes did not have to learn unintuitive rules about cause and effect. They simply observed that removing a supporting stick resulted in the lid going down.

(6)

“Line 138 When did the trial actually start; when the door opened, allowing the chimpanzees into the test room or when the first chimpanzee touched the first stick? According to Lines 425-426 and mentioned in several other spots, the trial started when the door between the rooms opened. In Lines 602-604 – is it correct that lid latency began when the trial began (when the door opened)?”

Yes, the trial started when the door opened. *Lid Latency* and *Stick Latency* begin at the start of the trial (i.e. the moment the door was opened). We have added the highlighted text to line 160 to make this clear:

*“We used the latency between the start of the **test (the moment the doors to the testing room opened)** and the moment of resource collapse as a measure of success in resource sustainability.”*

And to line 185, regarding *Stick Latency*:

*“Hypothesis 2 stated that **all sticks would be removed** from the common pool more quickly in the absence of the social dilemma (i.e. in the control condition). **This was measured in the latency between the start of the trial (the moment the door to the testing room opened) and the moment the last remaining stick was picked up.**”*

(7)

“Might it have made more sense to have lid latency begin when the first stick was touched? Did chimpanzees IMMEDIATELY head for and touch a stick when the door to the test chamber was opened? If so, it might be worthwhile to state this.”

It is not something we explicitly coded, but in almost all trials, at least one chimpanzee (but usually all of them) entered the testing room the second the door opened. Some of them rushed to take a stick as fast as possible, others did not.

Our perspective is that the task starts as soon as the chimpanzees enter the testing room, which is why *Lid Latency* and *Stick Latency* go from the moment the doors open. Hesitation to take a stick, even the first one, is relevant to the aims of the study, and so the time up until this point should be included in the measure.

(8)

“Lines 139-141 How did the chimpanzees know it was a control trial? Were the sticks positioned differently than during a test trial? Or was it simply that in the control condition, the lid was not present?”

The starting position of the sticks was always the same for dyads and the same for quartets, regardless of whether it was a test or control condition. The only difference between the test and control was that the lid was not present. This was highly conspicuous to the participants, as evidenced by the condition-difference (e.g. Figure 2 in the main text).

(9)

“Interobserver reliabilities were really, really high; much higher than one typically sees in behavioral research.”

Interobserver reliabilities were high because the judgements that the coders needed to make were unambiguous.

The first inter-rater reliability test did not involve interpreting animal behaviour at all. We used Pearsons R to correlate the video time code where the two coders judged the lid to be closed.

The subsequent Pearsons R tests correlated the two coders’ judgements of the time codes where each participant picked up and let go of each stick.

The Cohen’s kappa test assessed the agreement between the two coders’ counts of yoghurt-mouthfuls-eaten.

The scores we produced are not unusual for this type of behaviour coding, for example, in the SI of the Koomen and Herrmann (2018b) study that we cite, inter-rater reliabilities for collapse latencies are $r=1.0$ In Sánchez-Amato et al. (2024) report Cohen’s Kappa in the range of 0.93-0.99.

(10)

“Figure 3 is fairly difficult to interpret.”

We have attempted to improve the figure by increasing the line width and sending ribbon to the background.

(11)

“Might straws have been better than sticks? It would not have been necessary to remove straws, which eliminates the collapse aspect of the CPR, so probably not. But could the animals re-insert a stick, using it to stop the lid from coming down and closing off the resource during the approximately 10 seconds it took for the lid to close?”

There was nothing physically stopping the chimpanzees from re-inserting their sticks to prevent the 10 second collapse, but we found that they generally did not do this (illustrated by the strong correlation between last-stick-removal and collapse latency, see line 920 of the SI).

(12)

“The authors state on lines 471-472 that they did not observe the chimpanzees adopting turn-taking strategies. Do they have any thoughts about why this potentially quite successful strategy was not adopted? Similarly, if animal A in a quartet did not remove a stick in one trial, did animal B ‘sacrifice’ itself in the next trial of the same quartet and not remove a stick, allowing animal A to eat? Then animal C, etc.?”

Tasks that were designed specifically to elicit turn-based cooperation from chimpanzees find that they do not engage in turn-taking (Melis et al., 2016), though they may succeed in the task using other strategies, as was the case in our study.

While chimpanzees take turns in some sense, for example, they seem to “return the favour” after being groomed by a particular individual (Gomes & Boesch, 2011), it is thought that the proximate motivation for this is emotional bonds rather than calculated reciprocity. Melis et al. (2016) writes that the kind of turn-taking that the reviewer describes here requires inhibitory and future-planning capacities that chimpanzees rarely demonstrate.

We touch on this on line 179 and 1349, but we have not elaborated further, as it has been covered by other authors and is not the focus of this paper.

(13)

“Some of the reporting of the results (Lines 307-314), especially of 3-way interactions, gets confusing.”

We agree with the reviewer and have re-written much of the Results section to improve clarity.

In the section that the reviewer highlights here (line 913–940 in the revised manuscript), the texts now reads:

“Payoffs were measured by counts of scoops of yoghurt consumed. Our pre-registered hypothesis (H4) stated that payoffs would be less equal in dyads compared to quartets, especially in dyads with high Group Dominance Difference and low Group Social Tolerance.

This hypothesis was not supported. The GLMM (model 6) found group size to be negatively associated with payoff equality, meaning that payoffs were more unequal in quartets compared to dyads, with an estimate of -2.32 (95%CI[-3.52, -1.06]. However, trials that were sustained for longer tended to have lower payoff equality, with an estimated effect of -0.76

(95%CI[-0.92, -0.60]). This association was stronger in dyads, indicated by the positive interacting effect of Collapse Latency and Group Size (0.92, 95%CI[0.64, 1.18]). The interpretation of this is straightforward; as the participants did not share sticks or establish turn-taking, long dyadic trials will intrinsically be unequal, resulting in this strong interaction. If one participant is feeding, in order for the second player to feed, they must remove the second stick, triggering resource collapse. Therefore, in most instances, the longer the trial runs for, the more unequal it will become, creating a strong direct relationship between latency and equality. Despite this, payoffs were generally more unequal in quartets, but the strength of the association between payoff equality and latency was reduced.”

(14)

“Line 348 What does it mean to be “good” at the task? Not removing a stick, so that the others can eat?”

We have rephrased this for clarity (line 1146):

Before:

“Additionally, as success in this task can be achieved by at least one participant declining to remove a stick, and as some participants are more likely to fulfil this role than others, quartets may have an advantage over dyads simply because having more players increases the chances of an individual who is particularly “good” at the task being present. However, this would not explain why the dyads containing the particularly able individuals fail to reach collapse latencies as long as the best quartets.”

Changed to:

“Furthermore, success in this task can be achieved by just one participant declining to remove a stick, and some participants are more likely to fill this role than others. Quartets may therefore have an advantage over dyads simply because having more players increases the chances of having one of these players in the group.”

(15)

“Lines 529-535 A session contained 1-5 trials. How were trials separated from one another? Were the chimpanzees moved out of the testing room and put back in the waiting room? Did each trial begin with a full allocation of yoghurt?”

Thank you for catching this missing piece of information. We have added the following text to line 284:

“In a test-condition trial, the trial ended when the lid was collapsed, or when a cutoff was imposed (see SI section 2.5 for criteria). In a control-condition trial, the trial was over when all sticks were removed. As there was no collapse mechanism in the control-condition, after the

last stick was removed, the apes were left to freely eat the yoghurt for a minimum of one minute (though it was usually longer)."

And line 298:

"If a trial was the last of the day, the group would be released into their enclosure shortly after the trial finished. If they were to participate in another trial, they would be returned to the "waiting room", the apparatus would be refilled with yoghurt, returning the volume to approximately 1kg, and the procedure would be repeated."

(16)

"Do the authors find it surprising that there was no learning effect across sessions?"

A little! We hypothesised that latencies would increase over trials, such as reported in Sánchez-Amaro et al. (2019) but thought that they may alternatively decrease over trials, as Koomen and Herrmann (2018b) report, due to a ratcheting up of competition.

Instead we found that some groups increased in latency over test-condition trials, some decreased, and some had no pattern. Overall, it may be that individual-level learning effects are obscured by using a group-level outcome measure.

(17)

"The pre-experience photo is a little confusing. How many sticks are there in this photo?"

We apologise for the confusion. There are three sticks, as shown in the photo, however the description in the "stick position" column was incorrect.

We have now corrected this to state the three correct stick locations: "3,2 and 9,2 and 8,6".

.....

Reviewer #2 (Remarks to the Author):

Tolerance in chimpanzee groups leads to more sustainable resource use in a Common Pool Resource dilemma

General Comments

“This paper was genuinely a delight to read. The apparatus design is intriguing, the language of the manuscript is relatively jargon-free and accessible to non-specialists, and the group size comparison indeed adds significantly to our understanding of the CPR behavioural characteristics of chimpanzees. Minor spelling/grammar errors throughout. Minor to moderate revision suggestions are provided for each section, below. No major revision suggestions are provided. The contributions of this paper are significant, particularly in light of the findings of Koomen & Herrmann, 2018b. Namely, the following contributions are noteworthy and important for the field of comparative psychology: the clever apparatus design, the dyadic vs quartet comparison, and the findings on a dominance difference * tolerance interaction in quartets. I applaud the authors for the work.”

We thank the reviewer for the positive feedback and helpful suggestions. We are appreciative that they saw considerable merit in our work.

Introduction

(1)

“P4 Line 108-109 – three studies were identified but then only two are cited?”

Three studies were cited, but the wording was confusing. The manuscript previously stated:

“Including the previously described collective action study by Schneider et al. (2012), we have identified three captive experimental studies of chimpanzee on the topic of cooperation and social dilemmas that include more than 2 participants.” We then proceeded to cite Sánchez-Amaro et al. 2024 and Suchak et al. 2016.

For clarity, we have changed this to:

“**In addition to** the previously described collective action study by Schneider et al. (2012), we have identified **two** captive experimental studies of chimpanzee on the topic of cooperation and social dilemmas that include more than 2 participants” (line 126).

(2)

P4 line 123 – collapse mechanism of study 1 is not arbitrary, it is linked directly and visually with the amount of the resource that is available at any given time

We agree with the reviewer that “arbitrary” was not the correct choice of word here. We meant to convey that the collapse mechanism was difficult for the chimpanzees to learn, perhaps because it was unintuitive to them. We have removed “,or learn arbitrary rules” from the text (line 141 in revised manuscript).

(3)

P6 line 152 – citation should be 2018b, not c

Thank you for catching this mistake. We have changed this in the main text and removed the 2018c paper from the bibliography.

(4)

P6 line 167-169 – Hypothesis 2: clarify which outcomes are being referred to in relation to which interpretation.

We have removed “When interpreting the results of this experiment, there must be a distinction between these two outcomes, as one would suggest that natural cofeeding tolerance mediates CPR management, while the other suggests more complex strategic behaviour.”

And added on line 190:

“For example, if a group contained one participant who consistently waited for over two minutes before taking a stick, while the other group member(s) took sticks within ten seconds, we would want to know whether the slower participant was prevented from approaching by low cofeeding tolerance. In this case, their latencies for removing sticks should be no different to their latencies in the control condition. If their behaviour had a more strategic motivation, where they refrained from removing a stick to avoid resource collapse, we would expect to see shorter stick-removal latencies in the control compared to the test condition, as the participant would have no reason to not approach and remove a stick in the control.”

(5)

“P7 – Hypothesis 4: it’s unclear to me why payoffs are predicted to be more unequal in dyads than in quartets”

We expected that dyads would produce longer collapse latencies than quartets. We expected that this would be achieved through monopolisation by the dominant individual. We did not expect the chimpanzees to share one stick while taking turns to scoop the yoghurt.

Long dyad trials would therefore *have* to be highly unequal in payoffs. This is because one chimpanzee in the pair would be sat feeding from the resource for the entire trial, and as soon as the second chimpanzee removed the remaining stick, resource collapse would shortly follow. Therefore, the participant who monopolised the resource might achieve 60 scoops of yoghurt, while the other participant would only be able to eat 1 or 2 before the lid fully closed. Trials where both participants removed sticks quickly would produce low but more equal payoffs (both participants might be able to achieve between 1 and 3 scoops).

(6)

6.1 “Hypothesis 2 is generally unclear to me. Why would apes approach and extract more quickly from the control than the CPR dilemma?”

We realise that “approach and extract” is too a vague a description, as we intended it to mean something more like “approach and extract all extractable parts of the apparatus” rather than “first approach”. The way the participants interact with the last stick in the resource is the key difference between the test and the control.

We have changed the wording of hypothesis 2 to a more precise description of our prediction (though the prediction itself is effectively the same). In the previous manuscript hypothesis 2 stated:

“... participants would approach and extract from the common pool more quickly in the absence of the social dilemma (i.e. in the control condition).”

This has been changed to:

“all sticks would be removed from the common pool more quickly in the absence of the social dilemma (i.e. in the control condition).”

This can be seen on line 185, 542, and in the model descriptions in Table 1.

We rephrased text on line 188 from:

“If no difference was found in the time until all of the sticks were removed in the test versus the control condition, this may suggest that conservative use of the resource is not motivated by concern for the social dilemma, but instead reflects cofeeding tolerance in the group (DeTroy, 2021).”

Changing it to:

“The measure aimed to distinguish between conservative use of the resource resulting purely from low cofeeding tolerance (DeTroy, 2021), and conservative use resulting from the social dilemma.”

We have also added text relevant to this comment while addressing point 4 (line 190). We expect all sticks to be removed more quickly in the control condition because we expect the apes to, firstly, understand that removing all of the sticks will trigger resource collapse and, secondly, be motivated to not collapse the resource.

6.2 “Couldn’t the opposite prediction also follow logically from the scramble competition involved in the CPR dilemma?”

This comment may be resolved already by removing the word “approach” from Hypothesis 2, however we have elaborated specifically on the reviewer’s question relating to scramble competition:

Based on previous findings (Koomen & Herrmann 2018b) we, firstly, expected some groups of apes to successfully overcome the social dilemma and delay resource collapse. We did *not* expect them to develop turn-taking or stick-sharing strategies as humans might. Therefore, we expected that the most likely solution for them would be to leave one stick inside the pool to keep it open. Secondly, we expected more dominance-asymmetric and lower tolerance groups to leave one stick inside for longer as the more dominant apes monopolised the resource (also based on Koomen & Herrmann 2018b). Therefore, the feeding competition that emerges in the test condition is more closely modelled by contest competition than scramble competition, though, like most real-world scenarios, it’s really a combination of both (van Schaik & Noordwijk, 1988).

Scramble feeding may be a more accurate model for the control condition, as a motivated ape should “scramble” to take a stick as quickly as they can, before another participant claims it.

6.3 “Also, why is approach not actually measured, but only total consumption time (ie latency to last stick)? Wouldn’t latency to first stick be a more direct measure of approach while also being more comparable across conditions?”

As noted in the reply to comment 6.1, we have removed the word “approach” from hypothesis 2, as it was misleading.

We thank the reviewer for this interesting suggestion! After consideration, we maintain that last-stick latency is the more informative measure to compare between the test and the control condition.

We understand that the reviewer is suggesting that the chimpanzees may rush to take a stick more quickly in the test condition compared to the control condition if they are competing over access to the resource. Alternatively, it may be that knowledge that the resource will become inaccessible through over extraction slows even the first ape from taking a stick.

However, our experience was that participants were highly motivated to get the reward in both conditions. The yoghurt was a rare, high-value food, and the chimpanzees were generally very excited to receive it. It was also still possible in the control condition that another player might take more than one stick, creating a motivation to approach quickly. We think it is most likely that there would be no difference in first-stick latency between conditions, as the presence of the lid does not explicitly affect the ability of the first ape to acquire a stick.

Investigating first-stick-latency would require adding three more models to the paper (refitting models 1-3 with first-stick-latency rather than last-stick-latency). As the paper

already includes seven pre-registered GLMMs (one only reported in the SI) and 2 additional non-pre-registered models, we would rather avoid fitting even more analyses to the same dataset when there is no clear hypothesis.

6.4 “Latency to last stick in the control condition is constrained only by the groups’ feeding preferences while latency to last stick in the CPR condition is constrained by the groups’ feeding preferences interacting with the physical affordances of the dilemma. I’m not sure how this is a measure of the chimpanzees’ awareness of the dilemma.”

It is not intended as a measure of their awareness of the dilemma, but as a measure of their ability to sustain the resource in the test condition, in comparison to their preference in the absence of a social dilemma in the control. The difference in latency until last stick removal illustrates that the participants do not purely scramble to get the sticks as fast as possible, but modify their behaviour in response to the social dilemma.

6.5 This is a matter of personal opinion, but I no longer cite Hardin because of his white supremacist writings. One way I have found to discuss the ‘tragedy’ concept is to cite the work of others who cited him (.... As cited in X).

Thank you for the suggestion. We agree that citing Hardin is not necessary in this paper, so we have decided to remove the citations. This can be seen on line 38.

We have also changed the text on line 1319 from:

“Koomen and Herrmann (2018b) have compared this to privatisation of resources to prevent over-use, which was advocated by Hardin in the original The Tragedy of the Commons paper, where he writes that privatisation is unjust, but “injustice is preferable to total ruin”. However, while Hardin recommends top-down regulation and exclusion of those with less power and money, our study finds that the low payoffs may be voluntarily accepted by the chimpanzee with the most leverage, the dominant.”

Changing it to:

“Koomen and Herrmann (2018b) have compared this to how humans may privatise resources to prevent over-use, a management strategy that usually involves excluding those with less power and money. However, our study finds that the low payoffs may be voluntarily accepted by the chimpanzee with the most leverage, the dominant.”

Results

(7)

“Quartets sustained the CPR for longer latencies than the dyads – this is what one would predict from random use of the resource because with four sticks in comparison to two, one

would expect that collapse occurs less quickly when all sticks (2 vs 4) are taken at random intervals independently of one another.”

Thank you, this is correct. We have added text to the discussion (line 1122) to acknowledge this:

“If the sticks were removed at random intervals, independently of one another, it would be expected that collapse would happen later if four sticks were present, compared to two. This is likely reflected in the difference between the control conditions: all sticks are removed, on average 38.9 seconds later in quartets compared to dyads. However, the difference in this measure between group sizes in the test condition is much greater (112 seconds), suggesting that this is unlikely to fully explain the group size difference.”

(8)

“Figure 3 and the ad-hoc analysis looking at relative payoff of the dominant individual provide a fascinating look at the dominance:tolerance interaction in chimpanzee CPR behaviour; certainly sheds new light on the findings of Koomen & Herrmann, 2018 (chimpanzee CPR paper).”

Thank you for this positive comment.

Discussion

(9)

“P13 – “by placing chimpanzees in dyads, experimenters may actually curtail the cooperative abilities of chimpanzees..” – this is an important insight from this study: The finding that sustainability success differences are found between dyads and quartets (and the novel comparative apparatus) provide a significant step forward in our understanding of the CPR behaviours of our closest living relatives.”

Thank you.

(10)

“P14 line 384-385 – ‘facilitates’ suggests an active role in both paradigms. Suggested: “However, the two experiments involve a similar mechanism, where group benefits are interdependent upon individual decisions.””

We agree and have made this change. It can be found at line 1195 of the revised manuscript.

(11)

P16 penultimate paragraph – I would advise the authors against speculating on the lessons to be learned from a data set that does not yet exist.

We agree and have removed this paragraph.

Methods

(12)

Dominance is measured in this study via keeper reports of relative rank. Given that hypotheses are based on findings from Koomen & Herrmann, 2018b, it should be stated clearly that the quantitative measure of dominance difference in this study, and the binary rank differences in the dyadic study (Koomen & Herrmann, 2018b) differ.

(13)

Same comment for the measurement of group-level / dyad-level tolerance. In the present study this is a behavioural measure based on physical proximity while the tolerance measure in Koomen & Herrmann's dyads was a co-feeding measure based on duration of co-feeding from a shared pile of food. If direct comparisons are made between the findings of the three studies, it should be clearly stated that the social measures may be capturing a different quantitative impression of the relationships being compared. Perhaps the authors can also justify why they opted for these measures?

We have added the following text at line 313 to address this point:

“While our focus on dominance and tolerance draws from Koomen and Herrmann (2018b), we use different measures to investigate them. Koomen and Herrmann (2018b, Study 2) conducted a pre-test where their dyads of chimpanzees were given simultaneous access to a pile of carrot pieces. The individual that was able to consume more pieces was rated as the dominant, and the co-feeding tolerance measure was based on the latency of the subordinate individual to feed at the closest possible location to the dominant. It was not possible to conduct this type of pre-test at Leipzig Zoo, due to concerns that it could elicit aggression.”

(14)

“Were gaussian distribution GLMMs compared for fit against the gamma distribution family for the latency outcome data? Some CPR outcome data from human studies are more closely fit with gaussian distributions.”

We did not initially compare the fit with a Gaussian distribution, because visual assessment of the distribution of the data (for example, Figure 2), indicated that a Gaussian distribution was unlikely to be a better fit.

To check this, we have now refitted models 1 and 2 (that have *Collapse Latency* and *Last Stick Latency* as the respective outcomes) with Gaussian outcome distributions in order to conduct LOO comparisons. The LOO comparisons strongly favoured predictive power of the Gamma-log-link versions ($ELPD_{MODEL1} - ELPD_{NULL1} = -428.6, SE=22.7$; $ELPD_{NULL2} - ELPD_{MODEL2} = -710.0, SE=37.8$).

We then investigated whether log-transforming the outcome could make Gaussian a better fit. As this changes the outcome variable from *Collapse Latency* to $\log(\text{Collapse Latency})$, tests such as LOO and WAIC are not appropriate for comparing the fit of the models (<https://discourse.mc-stan.org/t/waic-and-loo-in-multivariate-models-with-different-response-variables/4007>).

Instead, we visually assessed the fit using posterior predictive checks (`pp_check()`). From the posterior predictive check plots, we could see that the log-transformation of the latency outcomes certainly improved the fit of the Gaussian models, however the original Gamma-distributed models are still the strongest fit.

Model 1

Model 2

Supplementary Information

(15)

“Looks very well organised and detailed.”

Thanks!

Additional comment:

(16)

“Given the nebulous nature of tolerance and the many ways it is operationalised in non-human primates (e.g., DeTroy, Haun, & van Leeuwen, 2022), given the fact that in this paper, tolerance is operationalised as time spent in proximity and it is being compared directly with findings from Koomen & Herrmann (2018), which operationalised co-feeding tolerance as time spent in proximity while co-feeding on a shared food source, AND given that the tolerance measure in the present study forms a central part of the main conclusions, I would caution the authors against making this claim in the title itself without a bit of clarification. Is, for example, the long-term Leipzig data on proximity more comparable with Engelmann & Herrmann (2016) operationalisation of "friendship" (SCI)? I'm not sure there is a straightforward answer here, because the measurement of relationship quality in non-human primates, as in humans, is a complex and sometimes fraught endeavour, but the title as it currently stands doesn't seem appropriate and the discussion on the use of 'tolerance' in the paper is insufficient. This does not, however, take away from the importance of the findings.”

We can see the reviewer's point. We have suggested a new title: *Chimpanzee groups achieve sustainable resource use in a Common Pool Resource dilemma*

Reviewer 3

(1)

“My main concern with the methods used is that while correlation at the group-level is accounted for through a random-effect, correlation at the individual-level is not accounted for. Specifically, note that the same individual is present in multiple groups. As such, one might expect groups with the same individual to perform similarly, if there was one individual for example that consistently ate more yogurt than the others. Only adjusting for group-level correlation would not capture this individual effect, since the models used treat groups as independent categories when they in fact may be correlated. To address this, a random effect would need to be added to every model for individual. This should be straightforward to add to the models, though care should be taken when interpreting

regression coefficients. If there is a concern with overparameterization, stronger priors could be considered.”

We thank the reviewer for catching this. We agree, and have incorporated this suggestion into our modelling.

As every group contains multiple individuals, we added four columns in the dataframes to hold one name of each player in the group, labelled *player 1-4*. We then used `mm()` (for multiple group memberships) in the `brms` R package to add individual-level random effects to the model.

Individual-level random effects were added to all models.

Models 1, 2 and 3

Model 1 compares *Collapse Latency* for groups of different sizes, resulting in exclusively NA values in the “player3” and “player4” name fields when group size was two. To handle this, in dyads, we cloned the name of player1 into the player3 field, and player2 into the player4 field.

For models 1 and 2, effect estimates were not strongly affected by this change. To save space in this response letter, a summary of effects that we report in the paper, and changes made by adding `mm()` are reported in table R.1. The full model outputs can be found in the SI. The main text has been updated with the new numbers.

In model 3, the *pd* of the *Group Dominance Difference: Group Social Tolerance* parameter drops from 94% to 91%. As 91% *pd* still represents weak-to-moderate evidence, and as this interacting effect was interpreted cautiously in the original text, we have not changed the interpretation.

The *k*-fold cross-validation comparisons for the new model structures did not change greatly. The values have been updated in the SI (lines 317-330).

Model	Predictor	Original model			Updated model		
		Estimate	l-95%CI	u-95%CI	Estimate	l-95%CI	u-95%CI
1	Group size	0.87	0.34	1.36	0.86	0.31	1.37
2	Condition	0.25	-0.07	0.56	0.26	-0.08	0.57
2	Dom	-0.18	-0.49	0.16	-0.17	-0.50	0.18
2	Tol	-0.15	-0.44	0.15	-0.15	-0.46	0.16
2	Dom:Tol	-0.10	-0.50	0.30	-0.10	-0.50	0.32
3	Condition	0.70	0.39	1.01	0.70	0.37	1.02
3	Tol	0.67	0.16	1.08	0.61	0.00	1.08
3	Dom:tol	-0.55	-1.15	0.16	-0.49	-1.16	0.28

Table R.1

Models 4 and 5

Models 4 and 5 explore whether *Lid Latency* is predicted by the proportion of the reward that is eaten by the highest-ranked individual in the group, and the interaction of this variable with *Group Dominance Difference* and *Group Social Tolerance*.

Without making any other changes to Model 4, this resulted in overparameterisation, as the reviewer anticipated. The *k*-fold cross-validation no longer favoured the full model over the null model. As a first step, we simplified the model by removing predictors of shape. Based on the `pp_check()` plots (kernel density, histograms, and ECDF), this was an improvement.

We then took the reviewer's suggestion of tightening priors, and changed the prior for the Intercept from `normal(2,2)` to `normal(5, 0.3)`. The other priors were not changed. This greatly improved the fit, based on the `pp_check()` plots. We re-ran the *k*-fold cross validation and found slight preference for the full model, but with a fairly wide SE ($ELPD_{MODEL4} - ELPD_{NULL4} = -4.3$, $SE=6.1$).

Model 5 took the same structure as model 4, but the prior for the intercept was left as `normal(2,2)`. The addition of `mm()` did not change the parameter estimates substantially.

		Original model			Updated model		
Model	Predictor	Estimate	l-95%CI	u-95%CI	Estimate	l-95%CI	u-95%CI
4	Proportion	-0.20	-0.41	0.01	-0.18	-0.37	0.03
4	Proportion:dom	-0.14	-0.40	0.12	-0.17	-0.43	0.08
4	Proportion:tol	-0.04	-0.27	0.19	-0.06	-0.28	0.18
5	Proportion	0.02	-0.06	0.09	0.04	-0.05	0.12

Model 6, 7 and 8

Random effects of individual were added to models 6, 7 and 8. Slight changes to effect sizes of predictors are summarised below. No results of model 7 were reported in the main text, so they are not summarised here, but the full output can be found in Table S13 of the SI.

Adding a random effect of individual increased the width of the 95%CI interval for the effect of tolerance in model 8, changing the probability of a negative effect from 96% to 93%, as this was reported tentatively in the original version, we have not changed the text, other than updating the values.

Model	Predictor	Original model			Updated model		
		Estimate	l-95%CI	u-95%CI	Estimate	l-95%CI	u-95%CI
6	Group size	-2.18	-3.43	-0.94	-2.32	-3.52	-1.06
6	Log(Collapse Latency)	-0.74	-0.91	-0.58	-0.76	-0.92	-0.60
8	Tolerance	-0.52	-0.92	0.07	-0.44	-0.95	0.17

(2)

“The authors state on line 592 that they conduct a simulation-based power analysis. If details on the power analysis were provided in the analysis plan linked to in Section 1 of the supplementary information, please also link to it here so that it is clear where one would go to replicate the power analysis. If details were not provided in the linked analysis plan, please add a section to the supplementary information giving details on the simulation you conducted so that this could be replicated.”

We have added the script to conduct the power analysis to the data repository, where the other analysis scripts and data are stored (<https://osf.io/q8ga6/>). We have now stated this on line 307 of the SI.

(3)

“For all hypotheses addressed using models in the Supplementary information, regression coefficients should be interpreted in context to make clear how exactly the highlighted rows in the summary tables correspond to the hypothesis in question. As an example, in Table S7 the row for cond cent is highlighted, but it is unclear what this has to do with collapse latencies compared between quartets and dyads.”

The coefficients are interpreted in the main text. We have removed the highlighting, tidied the tables, and insured consistent parameter labelling to reduce confusion.

(4)

“In the main body of the paper, HPD is stated on line 206, and this should be written out first as “highest posterior density.” Additionally, it might be worth justifying why the authors chose to use HPD intervals as opposed to a different form of credible interval, which is typically more common.”

We have changed the reporting to 95% Credible Interval rather than HPD. We introduce the term on line 535.

(5)

“On lines 608-609, the authors state that the prior for the intercept is set to $N(2,4)$ and that this represents the belief that lid latency must always be positive. A normal distribution with mean 2 and standard deviation 2 is not always positive, so please clarify what was meant by this.”

The reviewer is correct that a normal distribution with a mean of 2 and standard deviation of 2 is not strictly positive. It was intended to be a reasonable belief about the range of *Lid Latency*, with the Gamma response distribution ensuring that all predicted values were positive. We have updated the text on line 393 to reflect this:

“This moderately informative prior reflected the belief that *Lid Latency* must always be positive, but can vary flexibly”

Has been changed to:

“This moderately informative prior reflected a belief about the possible range of *Collapse Latency* that can vary flexibly, with the Gamma response distribution insuring that predicted values were positive.”

(6)

“The model formulas in the supplement appear to be copied from R code. It would be much more clear to instead write out model formulas in the form

$E[\text{outcome} \mid \text{predictor 1, predictor 2, ...}] = \beta_0 + \beta_1 \text{predictor 1} + \dots$

Alternatively, the tables of regression coefficients show what variables are in each model anyways, and so if the authors made the parameter names more informative in the table (i.e. not using R variable names) that may suffice.”

We have corrected inconsistencies in variable names in the output tables.

(7)

“Please reformat and check all tables in the Supplementary information, they appear to be copied from R console output, and should instead be formatted as tables in whatever form the journal requires (see Table S1, for example).”

Thank you, we have done this.

(8)

“There is a typo in the first sentence of Section 3.2 of the Supplementary Information: “hisection””

Thank you. We have corrected this.

(9)

“Section 3.3 of the Supplementary material discusses a calculation of correlation, but gives no results from this. Please state what the results were.”

This result was reported in the main text (line 918 of the revised manuscript). We have now added the following text to the SI (line 917).

“The two measures were found to be highly correlated ($r_s=0.956$, $p<0.001$, $n=103$).”

Dear Dr. Lui,

We are delighted that *Communication Psychology* is interested in publishing our work. Thank you to you and the reviewers for your help in improving the manuscript.

We have addressed the remaining concerns raised by the reviewers point by point below. The changes that were made to the manuscript are highlighted in *blue*.

While reviewing our models for their compliance with the editorial request table, we spotted two minor errors. Firstly, on line 687 of the revised MS we reported from model 4 that *Proportion* shifting from its lowest to highest value resulted in a 22 second increase in latency. This was recalculated as an 80 second increase. The impact of the highest ranker eating smaller amounts is therefore larger than we initially reported, but the direction is the same. The error came from using a non-centred version of *Proportion* in the calculation. This has now been corrected.

Secondly, in model 6, we originally did not centre the *Collapse Latency* predictor prior to fitting the model. This has now been changed, and we find that average payoff equality is higher in quartets compared to dyads, not lower. This does not affect the interpretation of the main findings of the study: the condition effect (models 2 and 3) and the group size effect (model 1). This correction can be seen in line 724 of the Results section.

Reviewer 1

“This is a revision of an interesting, well-written paper that explored a number of variables/hypotheses related to the Common Pool Resource (CPR) dilemma in nonhuman primates. In this study, captive chimpanzees living in Leipzig, Germany were the subjects. The authors devised an ingenious apparatus that allowed them to assess how various social variables might affect chimpanzees feeding from a device that, during the test condition, would ‘collapse’ if all of the animals attempted to use the sticks propping up the lid to feed from it. During the control condition, there was no lid, so the animals could remove and use the sticks with no adverse consequences. One of the major behavioral comparisons was between the responses of chimpanzee dyads (n=17) and quartets (n=7) drawn from two separate social groups. A large number of trials (708 over a 15-month period) were performed on a reasonable number of individuals (10 in one group and 5 in the other). The authors have successfully addressed most of the comments from the original submission. A small number of questions still remain, primarily for clarification purposes.

The manuscript is still quite long with a considerable amount of important supplementary information, especially concerning the experimental methods.”

(1)

“Did the chimpanzees ever show aggressive/frustration responses once the lid had

closed? The mesh barriers prevented them from ‘attacking’ the lid, but did they try to get more if yogurt was still present?”

In general, the chimpanzees did not respond aggressively to the lid closing. In most instances, they tried in vain to reopen the lid with their sticks, used their fingers to eat yoghurt that had been dropped in the vicinity, or simply walked away. We did not video-code behaviours after the end of the trial (i.e. lid close), so we do not have data aimed specifically at answering this question, however, during the testing phase, the task did not regularly elicit aggression or frustration from the participants. If it had, the procedure would have been modified for the apes’ welfare.

(2)

“Figure 1 is good.”

(3)

“How were non-study chimpanzees excluded from the ‘waiting’ room? The waiting room was not the same as the ‘colony’ room, correct?”

The testing room adjoined the main enclosure. The testing room could be split into compartments by opening and closing a series of internal doors. The waiting room was underneath the testing room, accessible through two doors in the floor.

To gather the participants, the keeper would open the doors between the main enclosure and the testing room and call the names of the relevant participants. Often, many chimpanzees entered the testing room (i.e. not only the participants). The chimpanzees respond to their own names, in addition to a number of commands such as “out”, “come”, and “move”. Using commands and food rewards to move chimpanzees between compartments, the participants could be separated from the non-participants. The non-participants would be rewarded for coming inside with a small piece of food, and then the door to the main enclosure would be re-opened for them to leave. They usually left at this point, but sometimes it took several attempts, where the door would be closed, the ape would be rewarded again, and then the door would be opened, accompanied by the “out” command.

The door to the waiting room was not opened until the non-participants had left, so they never had a chance to enter the waiting room.

The text has been updated on line 263 to clarify this:

“Participants were called into testing rooms that adjoin their main enclosure by a zoo keeper. Using commands and food rewards to move chimpanzees between compartments, the participants could be separated from the non-participants. When all of the non-participants had returned to the main enclosure, the groups of either two or four participants were moved

to a “waiting” room *connected* to the testing room. *The apparatus setup was then completed in the testing room. The trial began when the door between the waiting room and the testing room was opened. The apes could move freely between the two rooms throughout the test.*”

If a chimpanzee chose not to participate, how could s/he return to the rest of the social group?”

Participants could see the apparatus set up in the testing room from the main enclosure. If they did not want to participate, they would not enter the testing room. If the participants showed signs of distress during testing, the keeper would open the doors to the main enclosure, and the participants could immediately rejoin the main group. This is stated in line 235 of the manuscript:

“Testing was always voluntary, the individuals were not food or water deprived, and a trained animal caretaker did the handling. Returning to their social group was possible at any moment of the test, as well as terminating the test, given signs of discomfort or distress. We did not separate infants from their mothers.”

(4)

“What level of agreement existed among the four keepers that evaluated the animals’ relative dominance?”

Thank you for highlighting this missing piece of information. We have now run Kendall’s W to test agreement between raters and added the following text to the manuscript (line 301). We have also added the raw dominance data and script use for the Kendall’s W calculation to the OSF data repo.

*“Four keepers at the Wolfgang Köhler Primate Research Centre were asked to rank the 10 A-group participants and *three keepers were asked to rank five B-group participants in order of dominance (such as in Herrelko et al., 2012). Kendall’s coefficient of concordance found strong agreement between the four A-group raters (W=0.86, $\chi^2(8)=27.6$, $p=0.001$) and between the three B-group raters (W=0.96, $\chi^2(4)=11.5$, $p=0.022$). The mean of all ratings for an individual was taken as that individual’s relative dominance ranking.”**

The following text was added to the *Other statistical methods* section of the SI (line 477):

“Kendall’s coefficient of concordance was used to test agreement between dominance rank ratings, finding a strong agreement between the four raters of the 10 A-group chimpanzees (W=0.86, $\chi^2(8)=27.6$, $p=0.001$) and between the three raters of the 5 B-group chimpanzees (W=0.96, $\chi^2(4)=11.5$, $p=0.022$).”

(5)

“Do the authors have any data on the order in which chimpanzees used the sticks?”

This data can be extracted from the video coding. It was not included in the manuscript as we did not want to add more bulk to an already long paper, and we did not think that its inclusion improved the paper.

Did the ‘dominants’ routinely have the lowest latencies to stick use?”

Not necessarily (see plots below). The order the sticks were taken likely depended on many factors related to the individuals’ personalities and the different relationships that make up the test groups. This variation allowed us to run models 4 and 5 where we found that *if* the highest ranked apes receive lower payoffs, this is associated with longer trial latency.

These graphs have been added to lines 414 and 441 of the Supplementary Information with a short description:

“The highest ranked ape in the group did not necessarily take a stick first (see figure S1).”

Dominant's order of stick-taking by Group ID (quartets)

(6)

“What proportion of time was at least one chimpanzee ‘eating’ during the period prior to resource collapse?”

We are unsure what the “period prior to resource collapse” exactly refers to. The apes ate in 100% of trials. We recorded 10 minutes of only one ape or no apes using the resource in just eight out of 704 trials. This is reported in line 276 of the SI.

(7)

Lines 233-237 The descriptions of the dependent variables in this study are confusing.

From entry to lid collapse is the dependent measure for experimental trials, correct (Model 1)? Is entry to removal of the last stick (from the pool) the dependent measure for control trials? This should be clarified. If the dependent variable is the removal of the last stick from the pool, are these two dependent measures likely to be comparable? It seems that Models 2 and 3 used removal of the last stick as the dependent measure?

Thank you for highlighting that this was still unclear. Yes, *Last Stick Latency* was used as a proxy for *Collapse Latency* in condition comparisons as there is no true “collapse” in the control condition. In the test condition trials, removing the last stick almost always triggered the resource collapse, making these measures very strongly correlated. We have added text to this section to clarify this point (line 271):

“In a test-condition trial, the trial ended when the lid was collapsed, or when a cutoff was imposed (see SI section 2.5 for criteria). In a control-condition trial, the trial was over when all sticks were removed. As there was no collapse mechanism in the control condition, after the last stick was removed, the apes were left to freely eat the yoghurt for a minimum of one minute (though it was usually longer). The time that the last stick was removed was also recorded in the test condition, allowing trial latency comparisons between conditions. For analyses that only included test condition trials, the true collapse latency could be used. In the test condition, Last Stick Latency could be used to indicate the time of collapse as it was very strongly correlated with Collapse Latency ($r_s=0.98$, $p<0.001$, $n=374$).”

We have also inserted the following plot to the section 3.4 of the SI (other statistical methods) to further emphasise the relationship between these variables:

Figure S2

(8)

Lines 490-491 Might it be that dominant individuals are more likely to tolerate higher payoffs in the other members of the group?

In this section we are describing our a priori hypothesis before we ran Models 4 and 5. We cannot change the text here because it would mean changing the hypothesis after the results are known.

If we understand the reviewer correctly, they are suggesting that our finding that *Group Social Tolerance* was more strongly predictive of latency in low-dominance difference groups may be caused by the very high ranking chimpanzees being firstly, more tolerant of others and secondly, more likely to appear in high-dominance-difference groups.

However, this explanation is not satisfactory because *Group Dominance Difference* is relative. Two groups may have the same *Group Dominance Difference* score, but one could be composed of high-rankers and another of low-rankers.

(9)

SI Line 109 Do the authors have any thoughts concerning why performance did not improve over sessions (a learning effect)?

In the prior review round, the reviewer posed the same question:

“Do the authors find it surprising that there was no learning effect across sessions?”
(Comment 16).

We suggested that it may be that individual-level learning effects are obscured by using a group-level outcome measure. We have not added this to the manuscript as it is really just a speculation.

(10)

SI Lines 272-274 Wouldn't recording these latencies as 10 minutes make it seem as though the animals 'succeeded' at cooperating?

Eight out of 704 trials were cut off after 10 minutes of inactivity or active feeding of only one player. This was a welfare precaution to avoid keeping unwilling participants inside the testing room. However, their reluctance to use the resource was not necessarily due to disinterest in the task. It may result from them trying to avoid collapsing the resource or from not wanting to cofeed in proximity with the active feeder. In principle, these trials are no different to other high-latency trials and there is no justification to discard them. The nature of the task is that they cannot all be eating at the same time, and we are studying how this dilemma is resolved.

As we stress in the manuscript, human cooperation in this task would likely look quite different. We did not expect chimpanzees to take turns with the sticks, so at least one stick must remain in the pool, meaning that, as long as the resource is sustained, at least one player must be “sitting out” at any time.

Reviewer 2

“The authors have adequately addressed my concerns with the exception of the 4 points below. Once these are sufficiently addressed, I will recommend acceptance. I extend gratitude to the authors, in advance, for their contribution to the CPR literature.”

We thank the review for their careful evaluation and positive comments.

(1)

“Ms Line 3423. “does not require participants to undergo extensive training in order to participate (such as the collapse rule in Study 1 of Koomen and Herrmann, 2018b).”

Reviewer response: The implication here is that the participants were ‘trained’ to make sustainable choices which is not accurate, but this also precludes the possibility that the task demands rendered the DoG more challenging (i.e. stop sucking on delicious juice may be harder behaviourally than the choice to avoid picking up a stick to get delicious yoghurt). I would suggest: “to test this experimentally with simpler task demands than study 1 of koomen & herrmann” because in both paradigms it is impossible to tease apart the effect of DoG difficulties from comprehension difficulties and the use of ‘task demands’ remains conservatively ambivalent about this distinction.”

Thank you, this is an important point, and we had not fully considered the differing task demands aspect.

We did not mean to imply that the participants in (Koomen & Herrmann 2018b) were trained to make the sustainable choice. We were referring to how they learned the connection between the bobber entering the red danger zone, and the plug being released to cause resource collapse. The reviewer correctly points out that whether it was difficult for the participants to learn this causal relationship is conflated with task demands. We have removed the reference to K&H Study 1 from this section (line 125 in revised MS) because using it as an example here was unnecessary. Developing as comprehensible a task as possible was one of the aims of the project, so we feel it is important to reference this when describing the study aims. The text now reads:

*“...thirdly, to test this experimentally with an **intuitive** apparatus and **simple paradigm**. By developing an apparatus that the majority of participants could use without training, we hoped to **maximise the size and representativeness of the participant pool** (see SI section 2.2 for pre-experience details and comprehension criteria).” (Line 125)*

(2)

Ms Line 3595. “While our focus on dominance and tolerance dynamics draws from Koomen and Herrmann (2018b), we use different measures to investigate them. Koomen and Herrmann (2018b, Study 2) conducted a pre-test where their dyads of chimpanzees were given simultaneous access to a pile of carrot pieces. The individual that was able to consume more pieces was rated as the dominant, and the co-feeding tolerance measure was based on the latency of the subordinate individual to feed at the closest possible location to the dominant. It was not possible to conduct this type of pre-test at Leipzig Zoo, due to concerns that it could elicit aggression.”

Reviewer response: The framing of this paragraph might lead readers to conclude that unethical practices were conducted. I would urge the authors to simplify this paragraph by reducing the description of the methods differences to ‘co-feeding tolerance’ (and cite

some of the many previous studies who have operationalised tolerance this way) and 'proximity' tolerance (or some other term that differentiates it from co-feeding). And the authors can then say that due to institutional differences between the ape facilities, co-feeding tolerance measures were not possible.

Thank you for pointing this out. We have rephrased this section to address the point. The text now reads (line 313):

"While our focus on dominance and tolerance dynamics draws from Koomen and Herrmann (2018b), we use different measures to investigate them. Koomen and Herrmann (2018b, Study 2) established dominance and tolerance scores through a co-feeding pretest. Conducting this test was not possible in our zoo setting, so alternative methods were used. As a result, "tolerance" in this study is based on individuals' tendency to spend time in proximity to each other, not based on their cofeeding propensity in a cofeeding tolerance test (e.g. Koomen and Herrmann, 2018b; Melis et al., 2006)."

Additional Reviewer response: Line 4712. To further clarify this distinction in the discussion, I recommend adding a sentence like this around line 4712 -> "similarly, our tolerance measure was proximity-based and not co-feeding based as it was in koomen & herrmann. While this methodological difference may preclude strict comparison of the findings, we believe both tolerance methods measure something important about the quality of dyadic relationships that is distinct from hierarchical differences and generalisable across paradigms."

We have added text to the Limitations section (line 907):

" Similarly, our tolerance measure was based on proximity data to quantify association, while in Koomen and Herrmann (2018b) it was calculated from a co-feeding tolerance test. While this methodological difference may preclude strict comparison of the findings, we believe both approaches capture the quality of dyadic relationships, in a way that is distinct from hierarchical differences and generalisable across paradigms."

(3)

Reviewer comment. Pages 10 – 11 of the authors' Response to Reviewers states the below, but I don't see any of this clarification reflected in the manuscript itself. This needs to be clarified in the ms.

6.2 "Couldn't the opposite prediction also follow logically from the scramble competition involved in the CPR dilemma?"

This comment may be resolved already by removing the word "approach" from Hypothesis 2, however we have elaborated specifically on the reviewer's question relating to scramble competition:

Based on previous findings (Koomen & Herrmann 2018b) we, firstly, expected some groups of apes to successfully overcome the social dilemma and delay resource collapse.

We did not expect them to develop turn-taking or stick-sharing strategies as humans might. Therefore, we expected that the most likely solution for them would be to leave one stick inside the pool to keep it open. Secondly, we expected more dominance-asymmetric and lower tolerance groups to leave one stick inside for longer as the more dominant apes monopolised the resource (also based on Koomen & Herrmann 2018b). Therefore, the feeding competition that emerges in the test condition is more closely modelled by contest competition than scramble competition, though, like most real-world scenarios, it's really a combination of both (van Schaik & Noordwijk, 1988). Scramble feeding may be a more accurate model for the control condition, as a motivated ape should "scramble" to take a stick as quickly as they can, before another participant claims it.

We have added text to this paragraph (line 177) to make explicit why we hypothesised that *Last Stick Latency* would be shorter in the control compared to the test, and not the other way around.

"The measure aimed to distinguish between conservative use of the resource resulting purely from low cofeeding tolerance (DeTroy, 2021), and conservative use resulting from the social dilemma. For example, if a group contained one participant who consistently waited for over two minutes before taking a stick, while the other group member(s) took sticks within ten seconds, we would want to know whether the slower participant was prevented from approaching by low cofeeding tolerance. In this case, their latencies for removing sticks should be no different to their latencies in the control condition. If their behaviour had a more strategic motivation, where they refrained from removing a stick to avoid resource collapse, we would expect to see shorter stick-removal latencies in the control compared to the test condition, as the participant would have no reason to not approach and remove a stick in the control. We did not expect the chimpanzees to develop turn-taking or stick-sharing strategies as human might. Therefore, the most likely solution would be for them to leave one stick inside the pool to keep it open. The last stick removal latency was therefore of interest, as it was expected to trigger resource collapse."

(4)

Reviewer comment. Page 12 of the authors' Response to Reviewers states the below, but again from the authors' response it appears as though no clarification was added to the manuscript itself. This needs to be clarified in the ms.

6.4 "Latency to last stick in the control condition is constrained only by the groups' feeding preferences while latency to last stick in the CPR condition is constrained by the groups' feeding preferences interacting with the physical affordances of the dilemma. I'm not sure how this is a measure of the chimpanzees' awareness of the dilemma."

It is not intended as a measure of their awareness of the dilemma, but as a measure of their ability to sustain the resource in the test condition, in comparison to their preference in

**the
absence of a social dilemma in the control. The difference in latency until last stick
removal
illustrates that the participants do not purely scramble to get the sticks as fast as possible,
but modify their behaviour in response to the social dilemma.**

We have added text to line 779 to address this:

“Quartets of chimpanzee took on average 103% longer to remove all sticks from the CPR in the test condition, compared to the control. This indicates their ability to sustain the resource in social dilemma test condition, despite a tendency to remove the sticks more quickly in the non-social-dilemma condition.”

Reviewer 3

“I am satisfied with the author's revisions based on the original comments I had made. My main concern was with the random effect for individual, and since that was addressed appropriately, I think the statistical methods used are now suitable. I appreciate the response and updates to my additional comments as well.”

Thank you for your suggestions to improve the manuscript.